# STABILIZED NEURAL DYNAMICS FOR BEHAVIORAL DECODING VIA HIERARCHICAL DOMAIN ADAPTATION

## ABSTRACT

Brain-Computer Interfaces (BCI) have demonstrated significant potential in neural rehabilitation. However, the variability of non-stationary neural signals often leads to instabilities of behavioral decoding, posing critical obstacles to chronic applications. Domain adaptation technique offers a promising solution. Nonetheless, the existing direct adaptation within latent spaces could result in feature deviations. Therefore, developing a stable and efficient alignment framework is crucial for neural decoders. In this work, we find that dynamical latent features can be extracted from neural dynamics utilizing causal architectures. We also demonstrate that the process of self-consistent alignment can generate more stable latent features. Based on these insights, we propose a novel hierarchical domain adaptation (HDA) method for the alignment of dynamical latent features. Using Lyapunov theory, we further analytically validate the stability of dynamical features, which experimentally exhibit significant enhancements across various datasets. Our HDA approach effectively addresses the challenge of non-stationary neural signals, thereby potentially improving the reliability of BCIs.

## 1 INTRODUCTION

Brain-Computer Interfaces (BCI) offer a direct pathway for connecting the brain with external devices, demonstrating great potential in neural rehabilitation for people with paralysis (Collinger et al., 2018; Chaudhary et al., 2016; Willett et al., 2021; Metzger et al., 2023; Willett et al., 2023). Despite recent advances, one key challenge for BCIs is how to maintain stable performance, considering that the non-stationary neural recordings could vary across days (Perge et al., 2013; Wimalasena et al., 2020). The variability in neural signals could stem from various factors, such as environmental conditions (Santhanam et al., 2007), device degradation (Woeppel et al., 2021), physiological changes (Athalye et al., 2017) to foreign materials, and behavioral changes (Truccolo et al., 2008). Consequently, frequent recalibration of a BCI system is necessary to maintain its performance, leading to a critical barrier to chronic applications (Pandarinath & Bensmaia, 2022).

To alleviate the burden of recalibration, some studies aimed to develop automatic decoder adjustment approaches to cope with variability in neural signals(Wimalasena et al., 2020; Degenhart et al., 2020). One strategy is to align the neural signals across multiple days. These approaches allow neural decoders trained on one day to apply to another day directly. To achieve this, unsupervised domain adaptation (UDA) techniques have been employed to align the distributions of neural signals across different recording sessions. Existing UDA approaches for BCIs can be categorized into two types. The first type performs the distribution alignment in raw neural signal spaces (Farshchian et al., 2018; Ma et al., 2023). The second type aligns on the latent feature spaces and seeks for the spatio-temporal relationships in neural signals (Degenhart et al., 2020; Ju & Guan, 2022; Kobler et al., 2022; Cho et al., 2023; Jude et al., 2022).

Unfortunately, unlike conventional data such as images and videos, aligning neural signals is more challenging due to the inherently non-stationary nature of neural activities (Gallego et al., 2020). Directly aligning raw neural signals (Farshchian et al., 2018; Ma et al., 2023) or latent features (Karpowicz et al., 2022; Wang et al., 2023) may result in unstable features for decoding. Therefore, it is crucial to develop a stable and efficient alignment framework, thereby achieving a stable feature space for robust neural decoders.

Existing researches have shown that the brain executes various functions by converging towards attractors (Khona & Fiete, 2022), which are linked to dynamical stability in response to neural perturbations. Inspired by these observations, we propose that dynamical latent features can be extracted from neural dynamics utilizing causal architectures (Chen et al., 2024). Furthermore, we show that the process of self-consistent alignment within neural systems promotes the generation of more stable dynamical latent features. Building on these findings, we introduce a novel framework of hierarchical domain adaptation (HDA) that efficiently aligns dynamical latent features. Through validation grounded on Lyapunov theory (Angeli, 2002; Jiang & Wang, 2001), we analytically demonstrate that HDA enhances the dynamical stability of latent features, achieving stable neural decoders over extended periods. Experimental validation of HDA reveals significant improvements across various datasets. Our HDA approach effectively tackles the challenge of non-stationary neural signals, potentially improving the reliability of BCIs.

The main contributions of this paper are summarized as follows:

- **Causal Architectures**: Unlike existing UDA studies for BCI decoding, our research utilizes causal architectures (Chen et al., 2024) based on neural dynamics to extract latent features. Consequently, the cumulative final latent features (Gros, 2010) can be used for stable neural decoding. In addition, these dynamical features, derived from short-time windows, have the potential to meet the real-time operational requirements of BCIs.

- **Hierarchical Domain Adaptation**: We propose a novel framework for hierarchical domain adaptation (HDA) that enhances the dynamical stability of latent features, based on causal architectures. Our findings also indicate that a pre-controlled upper bound on latent feature deviations contributes to the dynamical stability using Lyapunov theory. A theoretical verification is provided in Section 3.4.

- **Experimental Validation**: We conduct extensive experiments on motor cortex datasets (Ma et al., 2023) to validate the superior performance of HDA compared with existing methods. Employing Lyapunov exponents, we have numerically verified that HDA enhances feature stability in non-stationary signals and effectively stabilizes behavioral decoding.

## 2 RELATED WORK

**Unsupervised Domain Adaptation** Unsupervised Domain Adaptation (UDA) aims to bridge the gap between labeled source domain(s) and unlabeled target domains by matching their distributions. Some studies have achieved by minimizing discrepancies based on specific metrics (Peng et al., 2019a; Sun et al., 2016; Sun & Saenko, 2016), such as the maximum mean discrepancy (Long et al., 2015; 2017a). Inspired by Generative Adversarial Networks, another line of research utilizes domain adversarial training to obtain domain-invariant features (Saito et al., 2018; Sankaranarayanan et al., 2018; Chen et al., 2020; Long et al., 2018). For instance, the widely-used Domain Adversarial Neural Network (DANN) (Ganin & Lempitsky, 2015; Ganin et al., 2016) optimizes feature extractors to generate domain-invariant features that confuse the trained domain classifier.

As mentioned in Section 1, for stabilizing BCI decoding over time, UDA-based alignment approaches have been utilized for unsupervised recalibrations within raw signal and latent feature spaces. Recently, consistent low-dimensional latent dynamics have been leveraged as the latent features for alignment (Karpowicz et al., 2022; Wang et al., 2023; Vermani et al., 2023; Pandarinath et al., 2018; Fang et al., 2023; Safaie et al., 2023). These latent dynamics, situated within the neural manifold (Degenhart et al., 2020; Gallego et al., 2017; Mitchell-Heggs et al., 2023), are assumed to provide a stable underlying representation of behavior.

Nevertheless, some intrinsic features, including low signal-to-noise ratios (Hu et al., 2010), frequently result in instabilities when attempting to align the high-dimensional raw neural signals (Wang et al., 2023). Meanwhile, alignment in latent spaces typically assumes its stability ensured by neural manifolds (Gallego et al., 2017; Mitchell-Heggs et al., 2023), lacking further consideration for the dynamical stability of latent features. For instance, NoMAD (Karpowicz et al., 2022) and the source-free and unsupervised alignment (Vermani et al., 2023) directly match latent dynamics, which may overlook the potential instability of the source domain's extracted latent features. s In contrast, our method proposes a novel hierarchical alignment based on causal architectures in neural dynamics. We demonstrate that the iterative process of self-consistent alignment can generate more stable latent

features. This optimization enhances the dynamical stability of latent features, thereby stabilizing the neural manifolds against stochastic perturbations.

**Representation Disentanglement** Representation Disentanglement has been used in UDA to learn domain-invariant features. In the field of computer vision, researchers have successfully applied this technique to disentangle semantic latent features for tasks such as cross-domain image classification (Cai et al., 2019; Lee et al., 2021; Lv et al., 2022) and video action recognition (Wei et al., 2023). Various methods have been explored, including reweighting source features for meta knowledge transfer (Wei et al., 2021) and utilizing deep adversarial autoencoders (Peng et al., 2019b). In the realm of time series analysis, researchers have disentangled semantically meaningful factors to control the shape of ECG signals (Li et al., 2022).

In neural data analysis, researchers have focused on building robust and generalizable representations using advanced networks such as transformers (Ye & Pandarinath, 2021; Liu et al., 2022; Le & Shlizerman, 2022). Recently, unified frameworks have been proposed to enable scalable representations across sessions and subjects (Azabou et al., 2023). To achieve this, representation disentanglement has been employed to understand how different neural populations encode diverse external stimuli and their intrinsic connections with observable behavioral variables. Supervised learning techniques, supported by auxiliary variables (Zhou & Wei, 2020), along with self-supervised learning approaches, such as contrastive learning (Cheng et al., 2020) or swapping (Liu et al., 2021), are used to identify latent variables that are directly related to observable variables.

Existing disentanglement approaches for neural data analysis primarily focus on learning domain-invariant representations directly, without performing distribution alignment. Such methods may work well when source domains contain sufficient samples of various sessions and tasks (Wang et al., 2020; Parnami & Lee, 2022). Considering that UDA often requires little data (Ma et al., 2023), they are more practical when less source data is available. In this study, we propose HDA, and leverage disentanglement techniques as a tool to decompose latent spaces for better distribution alignment. The use of UDA with disentanglement techniques to stabilize BCI decoding performance has not been explored.

## 3 METHODOLOGY

### 3.1 PROBLEM FORMULATION

We view the unsupervised recalibration of BCIs over time as a UDA problem (Long et al., 2017b). First, we define the domain $\mathcal{D}$ as follows: $\mathcal{D} = \{(x_1, y_1), (x_2, y_2), \ldots, (x_n, y_n)\}$, where $x_i(t)$ (for $t = 0, \ldots, w - 1$) represents a raw neural signal sample of window length $w$ (with $w$ being significantly shorter than the length of the entire trial) from one session, and $x_i(t) \in \mathbb{R}^m$. The behavioral label $y_i$ corresponds to the $(w - 1)$-th time step, and $y_i \in \mathbb{R}^d$. Moreover, we define $\mathbf{X}$, $\mathbf{Y}$ as the random variable representing neural signals $x_i$ and the corresponding $y_i$ from $\mathcal{D}$. Based on $\mathcal{D}$, we further define a labeled source domain $\mathcal{D}_S$ encompassing neural signals and labels from a single session: $\mathcal{D}_S = \{(x_1^S, y_1^S), \ldots, (x_{n_S}^S, y_{n_S}^S)\}$. Concurrently, the unlabeled target domain $\mathcal{D}_T$ includes signals from a separate session: $\mathcal{D}_T = \{x_1^T, \ldots, x_{n_T}^T\}$. For convenience, we define $\mathbf{X}^S$, $\mathbf{Y}^S$ as the random variable representing neural signals $x_i^S$ and the corresponding $y_i^S$ from domain $\mathcal{D}_S$, respectively. $\mathbf{X}^T$ denotes the random variable of original signals $x_i^T$ from domain $\mathcal{D}_T$. Due to various factors, the distribution mismatch between $\mathbf{X}^S$ and $\mathbf{X}^T$ prevents the direct application of a decoder trained on $\mathcal{D}_S$ to $\mathcal{D}_T$. Our goal is to maintain decoding performance in $\mathcal{D}_T$ by stabilizing the extracted latent features via HDA and ensuring a consistent mapping to the label space.

### 3.2 THE FRAMEWORK OF HIERARCHICAL DOMAIN ADAPTATION

To stabilize extracted latent features, we propose HDA based on the dynamical latent states, as shown in Fig. 1(a). Initially, we employ an unsupervised alignment strategy to align the raw signal distribution of $\mathcal{D}_T$ with that of $\mathcal{D}_S$. As outlined in Section 3.4, we found that this step effectively maintains an upper bound on latent feature deviations under control. The aligned neural signals are then provided as external inputs to a dynamical system for extracting dynamical latent dynamics.

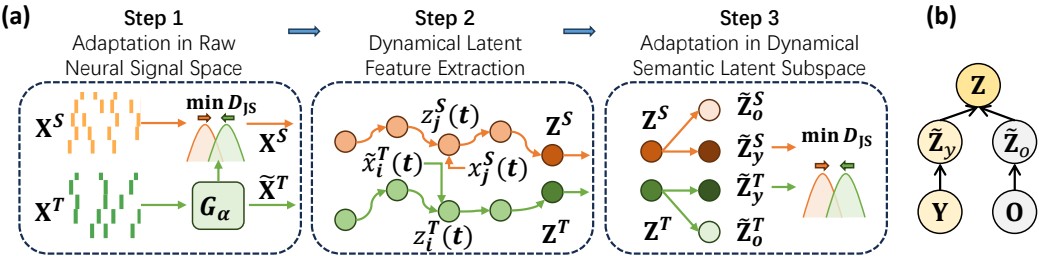

Figure 1: (a) The overall framework of HDA. (b) The model of generating $\mathbf{Z}$, which is controlled by the label variable $\mathbf{Y}$ and domain variable $\mathbf{O}$.

Subsequently, drawing inspiration from existing researches on learning interpretable and generalizable latent variables within neural signals (Zhou & Wei, 2020; Liu et al., 2021), we extract latent variables that are directly related to behavioral labels. These variables are then identified as the latent semantic features used for decoding. We further provide the dynamical systems with self-consistent alignment of these semantic features as feedback. To verify the stability of latent features, we employ Lyapunov stability (Angeli, 2002; Jiang & Wang, 2001) for a quantitative analysis of the system's stability. We have found that by optimizing parameters of the nonlinear dynamical system through HDA, the dynamical stability of latent features can be enhanced, thereby stabilizing neural decoders.

### 3.2.1 ADAPTATION IN RAW NEURAL SIGNAL SPACE

We begin by aligning the distribution of target signals with that of the source signals. The characteristic of spike signals is their capability to capture neural activities at the neuron level (Buzsáki, 2004), which ensures the sparse dependencies among different channels (Chen et al., 2010; Bighamian et al., 2019). In comparison to latent feature spaces, where spatio-temporal dependencies are more intricate, the raw neural signal spaces may offer a more direct causal relationship that facilitates the identification of units affected by drifts. This identification may help to align the distribution of the raw signals and forms the basis of our approach to enhance feature stability. Additionally, aligning input raw neural signals prior to optimizing system parameters enables a more statistically consistent input representation for our dynamical system. This approach contrasts with those that apply shared-parameter encoders to derive latent features directly from the original signals across source and target domains (Wei et al., 2023; 2021).

Considering the unique properties of biological systems, we found that probability densities based on individual samples may be more accurate to measure distribution discrepancies. This is because sufficient statistics, such as the mean, typically characterize the collective properties of random variables. However, the characteristics of individual samples are often more critical due to the common presence of outliers in biological systems (Gomez-Ramirez & Sanz, 2013). Therefore, we chose the $f$-divergence, which is based on probability density functions, to measure the discrepancy between distributions. However, since $f$-divergence is difficult to compute directly, we employed GANs to implement alignment based on $f$-divergences in an indirect manner. Given that naive GANs often suffer from training instability, we used LSGANs (Mao et al., 2017) based on the $\chi^2$ divergence, which is a specific case of $f$-divergence. The benefits of alignment based on $f$-divergences are demonstrated in Fig. 7(b).

Specifically, our optimization objective is to identify a nonlinear transformation based on $G$ that minimizes the $\chi^2$ divergence between the distribution of $\mathbf{X}^S$ and $G(\mathbf{X}^T)$, denoted as $D_{\chi^2}\left(p(\mathbf{X}^S)\|p(G(\mathbf{X}^T))\right)$. Here, this nonlinear transformation is implemented via a generator $G_\alpha$ from LSGAN, which is more stable than vanilla GANs, with parameters $\alpha$. As mentioned in (Mao et al., 2017), it is implemented by alternately training the generator $G_\alpha$, and a discriminator $D_\beta$ (with parameters $\beta$) via a min-min optimization based on the least-square loss functions $\mathcal{L}_{\mathrm{lsd}}(\beta)$ and $\mathcal{L}_{\mathrm{lsg}}(\alpha)$, respectively:

$$\min_\beta \left\{ \mathbb{E}_{\mathbf{X}^S \sim \mathcal{D}_S}\left[ (D_\beta(\mathbf{X}^S) - 1)^2 \right] + \mathbb{E}_{\mathbf{X}^T \sim \mathcal{D}_T}\left[ \left(D_\beta(G_\alpha(\mathbf{X}^T))\right)^2 \right] \right\} = \{\mathcal{L}_{\mathrm{lsd}}(\beta)\},$$

$$\min_\alpha \mathbb{E}_{\mathbf{X}^T \sim \mathcal{D}_T}\left[ \left(D_\beta(G_\alpha(\mathbf{X}^T)) - 1\right)^2 \right] = \{\mathcal{L}_{\mathrm{lsg}}(\alpha)\}. \tag{1}$$

After the adaptation within the original data space, we proceed with the extraction and alignment of dynamical latent features using $\mathbf{X}^{\tilde{S}}$ and $G_\alpha(\mathbf{X}^T)$.

### 3.2.2 DYNAMICAL LATENT FEATURE EXTRACTION

To achieve real-time extraction of latent features for decoding, we employ a causal architecture (Chen et al., 2024) based on nonlinear dynamical systems to extract latent dynamics from the raw neural signals. The initial latent state of this system is based on signals that were recorded at the onset of time windows, rather than a posterior distribution of the entire trial (Karpowicz et al., 2022; Wang et al., 2023; Vermani et al., 2023). The latent state evolution is jointly driven by the current time step's externally input signal and the latent state from the previous step through a nonlinear transformation.

We utilize $x_i(t)$ as the external input to the dynamical system at time $t$, and the corresponding low-dimensional latent state is denoted as $z_i(t) \in \mathbb{R}^k$ (where $k < m$). The initial latent state is determined by the function $g\colon \mathbb{R}^m \to \mathbb{R}^k$, and the evolution of the latent state is determined by the nonlinear function $f\colon (\mathbb{R}^k \times \mathbb{R}^m) \to \mathbb{R}^k$. Thus, the initial state and the subsequent evolution of the nonlinear dynamical system are characterized by the following equations:

$$z_i(0) = g(x_i(0)), \quad z_i(t) = f(z_i(t-1), x_i(t)) \, (t = 1, \ldots, w-1). \tag{2}$$

Considering the cumulative effect (Gros, 2010), we utilize the latent state $z_i(w-1)$ at the final time step to represent the dynamical latent feature corresponding to $x_i$, which is further transformed for decoding $y_i$. Specifically, we construct this nonlinear dynamical system using an LSTM-based (Hochreiter & Schmidhuber, 1997; D'Amico et al., 2023) network $E_\gamma$ (with parameters $\gamma$). The input of $E_\gamma$ is $x_i$, and the cell state is regarded as the latent state $z_i$. The cell state at the final time step, denoted as $z_i(w-1)$, is viewed as the output dynamical latent feature of $E_\gamma$: $z_i(w-1) = E_\gamma(x_i)$.

### 3.2.3 ADAPTATION IN DYNAMICAL SEMANTIC LATENT SUBSPACE

When performing a specific task, the brain processes a wide range of information, including perception, decision-making, environmental cues, feedback, and more. For instance, task-irrelevant perceptual information and environmental feedback are also encoded within the latent dynamics. By decomposing these latent spaces to remove irrelevant components, we aim to reduce variability within the latent dynamics, thereby improving alignment of latent spaces. Specifically, inspired by previous studies mentioned in Section 2 and the high parallelism (Wässle, 2004) of brains, we hypothesize that the drifts of dynamical latent features in the target domain primarily stem from latent variables that are loosely connected to observable behavioral variables. Based on this hypothesis, we believe that constructing a semantic subspace, by extracting components of the latent space that are directly related to behavioral variables, can effectively reduce the drift of neural population dynamics. Furthermore, we have found that, compared to directly aligning latent features, performing alignment only within the semantic subspace can provide the dynamical system with a more efficient self-consistent feedback, guiding it to obtain more stable latent dynamics.

**Decomposition of the Dynamical Latent Space** Based on the above hypothesis, the generation of dynamical latent features $z_i(w-1)$ extracted by $E_\gamma$, denoted by the random variable $\mathbf{Z}$, is assumed to be governed by two independent variables: the domain variable $\mathbf{O}$ and the observed behavioral variable $\mathbf{Y}$. These two variables form two independent subspaces. As depicted in Fig. 1(b), we decompose the dynamical latent features $\mathbf{Z}$ into two independent components based on these two variables: one part directly encodes the semantic information, and the other part directly encodes the domain information.

Existing studies (Zhou & Wei, 2020; Liu et al., 2021; Cai et al., 2019; Wei et al., 2023) have shown that Variational Autoencoders (VAE) can solve for latent feature subspaces determined by different variables. Here, we first use the VAE's encoder to transform original dynamical features $\mathbf{Z}$ into latent features $\tilde{\mathbf{Z}}$ ($\tilde{\mathbf{Z}} \in \mathbb{R}^{\tilde{k}}$) that follow a pre-defined Gaussian distribution. Then, we divide $\tilde{\mathbf{Z}}$ into two independent parts. The first $\tilde{k}_y$ dimensions represent components directly governed by $\mathbf{Y}$, denoted as the semantic latent features $\tilde{\mathbf{Z}}_y$ ($\tilde{\mathbf{Z}}_y \in \mathbb{R}^{\tilde{k}_y}$). The remaining $\tilde{k}_o$ ($\tilde{k}_o = \tilde{k} - \tilde{k}_y$) dimensions represent components directly governed by $\mathbf{O}$, denoted as $\tilde{\mathbf{Z}}_o$ ($\tilde{\mathbf{Z}}_o \in \mathbb{R}^{\tilde{k}_o}$): $\tilde{\mathbf{Z}} = [\tilde{\mathbf{Z}}_y, \tilde{\mathbf{Z}}_o]$. Finally, $\mathbf{Z}$ is reconstructed by the VAE's decoder using both $\tilde{\mathbf{Z}}_y$ and $\tilde{\mathbf{Z}}_o$. The Evidence Lower Bound (ELBO) loss function (Kingma & Welling, 2013) is further utilized to enforce independence (Higgins et al., 2017;

Burgess et al., 2018; Higgins et al., 2018) and reconstruction constraints on the decomposed latent subspaces after transformation. The VAE's encoder, denoted as $Q_\phi$ (with parameters $\phi$), estimates the posterior distribution $q_\phi(\tilde{\mathbf{Z}}|\mathbf{Z})$. The prior distribution of the transformed latent features $\tilde{\mathbf{Z}}$, denoted as $p_z(\tilde{\mathbf{Z}})$ is set to a multivariate Gaussian distribution by convention: $p_z(\tilde{\mathbf{Z}}) \sim \mathcal{N}(0, \mathbf{I})$. The VAE's decoder, denoted as $R_\theta$ (with parameters $\theta$), is used to estimate $p_\theta(\mathbf{Z}|\tilde{\mathbf{Z}})$. The ELBO can then be expressed as follows:

$$\log p(\mathbf{Z}) \geqslant \mathbb{E}_{\tilde{\mathbf{Z}} \sim q_\phi(\tilde{\mathbf{Z}}|\mathbf{Z})}[p_\theta(\mathbf{Z}|\tilde{\mathbf{Z}})] - D_{\mathrm{KL}}(q_\phi(\tilde{\mathbf{Z}}|\mathbf{Z})||p_z(\tilde{\mathbf{Z}})) = -\mathcal{L}_{vae}(\theta, \phi, \gamma), \mathbf{Z} = E_\gamma(\mathbf{X}). \quad (3)$$

Here, $D_{\mathrm{KL}}$ represents the Kullback-Leibler (KL) divergence, for which a closed-form solution can be directly provided for Gaussian distributions. Minimizing the divergence based on a Gaussian distribution with the zero covariance enforces the independence of decomposed subspaces after transformation, consistent with the previous hypothesis. We maximize $\mathbb{E}_{\tilde{\mathbf{Z}} \sim q_\phi(\tilde{\mathbf{Z}}|\mathbf{Z})}[p_\theta(\mathbf{Z}|\tilde{\mathbf{Z}})]$ by minimizing the reconstructed Mean Squared Error (MSE) loss. Finally, we achieve the maximization of ELBO by minimizing $\mathcal{L}_{vae}$.

**Further Constraints on Dynamical Semantic Latent Subspace** $\mathcal{L}_{vae}$ is not sufficient to ensure the encoding information of $\tilde{\mathbf{Z}}_y$ and $\tilde{\mathbf{Z}}_o$ as hypothesized. Therefore, drawing on related work (Cai et al., 2019; Wei et al., 2023), we introduce additional terms to constrain the information encoded within latent subspaces. Let $\mathbf{Z}^S$ denote the random variable representing dynamical latent features from $\mathcal{D}_S$. The corresponding latent features transformed by $Q_\phi$ are $\tilde{\mathbf{Z}}^S$, which are further decomposed into semantic components $\tilde{\mathbf{Z}}_y^S$ via corresponding parameters $\phi_y$ of $Q_\phi$, and domain-related components $\tilde{\mathbf{Z}}_d^S$ with corresponding parameters $\phi_d$. The semantic latent features $\tilde{\mathbf{Z}}_y^S$ are used to decode the behavioral labels $\mathbf{Y}^S$. Similarly, for $\mathcal{D}_T$, the dynamical latent features extracted from the aligned neural signals $G_\alpha(\mathbf{X}^T)$ are $\mathbf{Z}^T$, which are decomposed into semantic components $\tilde{\mathbf{Z}}_y^T$ and domain-related components $\tilde{\mathbf{Z}}_o^T$.

First, to ensure that $\tilde{\mathbf{Z}}_y^S$ and $\tilde{\mathbf{Z}}_y^T$ directly encode semantic information without the effect from domain variables, we optimize the decoding performance of semantic features and minimize distribution discrepancies between $\tilde{\mathbf{Z}}_y^S$ and $\tilde{\mathbf{Z}}_y^T$. Specifically, we use $\mathbf{Y}^S$ and $\tilde{\mathbf{Z}}_y^S$ for supervised training of a linear decoder $C_\eta$ (with parameters $\eta$), and measure the decoding performance of $\tilde{\mathbf{Z}}_y^S$ using the loss function $\mathcal{L}_y$: $\mathcal{L}_y(\gamma, \phi_y, \eta) = \|\mathbf{Y}^S - C_\eta(\tilde{\mathbf{Z}}_y^S)\|_2$, where $\tilde{\mathbf{Z}}_y^S = Q_{\phi_y}(\mathbf{Z}^S) = Q_{\phi_y}(E_\gamma(\mathbf{X}^S))$. Meanwhile, we minimize the conditional distribution discrepancy between $\tilde{\mathbf{Z}}_y^S$ and $\tilde{\mathbf{Z}}_y^T$ using the $\chi^2$ divergence $D_{\chi^2}\left(p(\tilde{\mathbf{Z}}_y^S|\mathbf{X}^S) \| p(\tilde{\mathbf{Z}}_y^T|G_\alpha(\mathbf{X}^T))\right)$. Similar to the raw signal alignment, we alternately optimize $\gamma, \phi_y$, and the discriminator's ($D_{\beta_y}^y$) parameters $\beta_y$ based on the loss function $\mathcal{L}_{bd}(\beta_y)$, and $\mathcal{L}_{bg}(\gamma, \phi_y)$ formulated as follows:

$$\min_{\beta_y} \left\{ \mathbb{E}_{\mathbf{X}^S \sim \mathcal{D}_S}\left[(D_{\beta_y}^y(Q_{\phi_y}(E_\gamma(\mathbf{X}^S))) - 1)^2\right] + \mathbb{E}_{\mathbf{X}^T \sim \mathcal{D}_T}\left[\left(D_{\beta_y}^y(Q_{\phi_y}(E_\gamma(G_\alpha(\mathbf{X}^T))))\right)^2\right] \right\}$$

$$\min_{\gamma, \phi_y} \left\{ \mathbb{E}_{\mathbf{X}^S \sim \mathcal{D}_S}\left[(D_{\beta_y}^y(Q_{\phi_y}(E_\gamma(\mathbf{X}^S))) - 1)^2\right] + \mathbb{E}_{\mathbf{X}^T \sim \mathcal{D}_T}\left[\left(D_{\beta_y}^y(Q_{\phi_y}(E_\gamma(G_\alpha(\mathbf{X}^T)))) - 1\right)^2\right] \right\}$$

$$(4)$$

Secondly, considering the linearity of $C_\eta$ and the independence constraint between the decomposed subspaces, $C_\eta$ could not work well with $\tilde{\mathbf{Z}}_o^S$ and $\tilde{\mathbf{Z}}_o^T$. Therefore, to ensure that $\tilde{\mathbf{Z}}_o^S$ and $\tilde{\mathbf{Z}}_o^T$ directly encode the domain information, we only constrain the domain relevance of $\tilde{\mathbf{Z}}_o^S$ and $\tilde{\mathbf{Z}}_o^T$. Here, we maximize the $\chi^2$ divergence between the conditional distribution of $\tilde{\mathbf{Z}}_o^S$ and $\tilde{\mathbf{Z}}_o^T$, represented as $D_{\chi^2}\left(p(\tilde{\mathbf{Z}}_o^S|\mathbf{X}^S) \| p(\tilde{\mathbf{Z}}_o^T|G_\alpha(\mathbf{X}^T))\right)$. Similarly, we alternately optimize the parameters $\gamma, \phi_o$, and the discriminator's ($D_{\beta_o}^o$) parameters $\beta_o$ based on minimizing the loss function $\mathcal{L}_{od}(\beta_o)$, maximizing $\mathcal{L}_{og}(\gamma, \phi_o)$ as defined in Eq. (4).

## 3.3 OVERALL LEARNING ALGORITHM

During the training phase, we initially optimize $G_\alpha$ and $D_\beta$ alternately based on $\mathcal{L}_{lsd}$ and $\mathcal{L}_{lsg}$. This step yields the aligned target neural signals, denoted as $G_\alpha(\mathbf{X}^T)$. For alignment within semantic

---

**Algorithm 1** Hierarchical Domain Adaptation

---

**Input:** source domain $\mathcal{D}_S$; target domain $\mathcal{D}_T$;
**Output:** signal aligner $G_\alpha$; latent dynamic extractor $E_\gamma$; VAE's encoder $Q_\phi$; linear decoder $C_\eta$
Initialize $G_\alpha, D_\beta, E_\gamma, Q_\phi, R_\theta, D_{\beta_y}^{y}, D_{\beta_o}^{o}$
**Adaptation in Raw Neural Signal Space**:
Optimize $G_\alpha$ and $D_\beta$ alternately based on $\mathcal{L}_{\text{lsg}}(\alpha)$ and $\mathcal{L}_{\text{lsd}}(\beta)$;
**Adaptation in Dynamical Semantic Latent Subspace**:
**for** $iter = 1$ **to** $n_{iter}$ **do**
   Sample mini-batch from $\mathcal{D}_S$ and $\mathcal{D}_T$;
   Update $D_{\beta_y}^{y}$ by $\mathcal{L}_{bd}(\beta_y)$; Update $D_{\beta_o}^{o}$ by $\mathcal{L}_{od}(\beta_o)$;
   Update $E_\gamma, Q_\phi, R_\theta, C_\eta$ by $\mathcal{L}_{total}(\gamma, \phi, \theta, \eta)$
   $(\mathcal{L}_{total}(\gamma, \phi, \theta, \eta) = \mathcal{L}_{vae}(\gamma, \phi, \theta) + \lambda_y \mathcal{L}_y(\gamma, \phi_y, \eta) + \lambda_b \mathcal{L}_{bg}(\gamma, \phi_y) - \lambda_o \mathcal{L}_{og}(\gamma, \phi_o))$;
**end for**
**return** $G_\alpha, E_\gamma, Q_\phi, C_\eta$.

---

subspaces, we proceed to train the feature extractor $E_\gamma$ based on dynamical systems, the VAE's encoder $Q_\phi$ and decoder $R_\theta$, and the linear decoder $C_\eta$. The training is guided by a combined loss function $\mathcal{L}_{total}$: $\mathcal{L}_{total} = \mathcal{L}_{vae} + \lambda_y \mathcal{L}_y + \lambda_b \mathcal{L}_{bg} - \lambda_o \mathcal{L}_{og}$, where $\lambda_y, \lambda_b$, and $\lambda_o$ serve as weighting factors for the respective losses. Concurrently, the discriminators $D_{\beta_y}^{y}$ and $D_{\beta_o}^{o}$ are trained based on $\mathcal{L}_{bd}$ and $\mathcal{L}_{od}$, respectively. A detailed description of the training procedure is presented in Algorithm 1.

### 3.4 VERIFICATION OF DYNAMICAL FEATURE STABILITY

Here, we propose a novel way to measure feature stability grounded in Lyapunov theory. First of all, for any two hidden states $z_i(t)$ and $z_j(t)$, the system is stable (Agrachev et al., 2008) if there exist functions $\beta(\|z\|, t)$ and $\gamma(\|x\|)$. For $t \geqslant 1$, the following inequality holds:

$$\|z_i(t) - z_j(t)\| = \|z_i(t, z_i(0), x_i(t)) - z_j(t, z_j(0), x_j(t))\| \leqslant \beta(\|z_i(0) - z_j(0)\|, t) + \gamma(\|x_i(t) - x_j(t)\|_\infty). \tag{5}$$

Furthermore, the stability defined above can be determined using a Lyapunov function $V(z)$. Given an equilibrium point $z^*$ of the system, the following equations are satisfied: (1) $V(z^*) = 0$, (2) $\dot{V}(z^*) = 0$, (3) $V(z) > 0$ for all $z \neq z^*$, (4) $\dot{V}(z) < 0$ for all $z \neq z^*$. It is known that $V(z) = \frac{1}{2}z^T z$ is one of the functions that meet the conditions. However, directly calculating complex $V(z)$ can be difficult. Therefore, we used the method based on (Wolf et al., 1985) to estimate the stability of $z(t)$ based on the maximum Lyapunov exponent (MLE). The maximum Lyapunov exponent $\lambda$ can be defined based on the latent state $z_i(t)$ as follows: $\lambda = \lim_{t \to \infty} \lim_{|\delta z_i(0)| \to 0} \frac{1}{t} \ln \frac{|\delta z_i(t)|}{|\delta z_i(0)|}$. A non-positive MLE often indicates the stability of dynamical systems, achieving stable dynamical latent features (Wolf et al., 1985). Here, the MLE $\lambda$ of $z_i$ is estimated to evaluate the stability of dynamical latent features extracted from $\mathcal{D}_S$ and $\mathcal{D}_T$ after adaptation. The detailed calculation of $\lambda$ and the theoretical explanation of how pre-alignment enhances stability are provided in Appendix A.2.3.

## 4 EXPERIMENTS AND RESULTS

### 4.1 EXPERIMENTAL SETUP

**Datasets** We utilized three distinct datasets of extracellular neural recordings obtained from the primary motor cortex (M1) of non-human primates (Ma et al., 2023), as outlined below. More detailed dataset descriptions can be found in Appendix B.1.
**Random-Target (RT-M)**. Monkey M was trained to move the cursor into a sequence of three randomly located targets on the screen within 2.0s after viewing.
**Center-Out Reaching (CO-C&CO-M)**. Monkeys C and M were trained to use an upright handle to aim for one of eight randomized targets upon receiving an auditory cue.
**Data Preprocess and Split** For all datasets, we extracted trials from 'gocue time' to 'trial end'. The data was then timestamped and smoothed with a Gaussian kernel to estimate firing rate over 50 ms

bins. We utilized the session labeled with 2D cursor velocity recorded on the Day 0 as $\mathcal{D}_S$ for training. As for training $\mathcal{D}_T$, we used 80% trials of the unlabeled session collected on another day. As for tests, we employed the remaining 20% trials from this session.

**Evaluation Metric** The deviation between decoded and actual cursor velocity is quantified using the $R^2$ score. All results presented below are averaged on five distinct random seeds. Further experimental details and settings are elaborated in Appendix B.

Table 1: Comparison of average $R^2$ scores (%) in cross-session velocity decoding

| Data | Session | LSTM | Cebra | DAF | ERDiff | NoMAD | Cycle-GAN | HDA | retrain |
|------|---------|------|-------|-----|--------|-------|-----------|-----|---------|
| CO-M | Day 0 | $74.18_{+4.90}$ | $\mathbf{79.24_{+4.90}}$ | $75.24_{+2.22}$ | $76.31_{+3.62}$ | $58.29_{+3.38}$ | $64.99_{+1.17}$ | $70.86_{+6.13}$ | $78.38_{+1.37}$ |
| | Day 8 | $-118.53_{+98.70}$ | $-51.92_{+98.70}$ | $-0.01_{+0.03}$ | $-75.33_{+83.37}$ | $57.86_{+2.25}$ | $71.30_{+1.46}$ | $\mathbf{76.84_{+1.19}}$ | $85.92_{+1.27}$ |
| | Day 14 | $-63.85_{+19.96}$ | $-1.77_{+19.96}$ | $-0.08_{+0.05}$ | $-102.82_{+34.63}$ | $63.45_{+1.41}$ | $67.05_{+1.36}$ | $\mathbf{71.70_{+1.50}}$ | $82.92_{+0.65}$ |
| | Day 15 | $-712.91_{+316.04}$ | $-83.24_{+316.04}$ | $0.03_{+0.03}$ | $-66.76_{+86.32}$ | $57.92_{+0.69}$ | $64.83_{+1.77}$ | $\mathbf{75.53_{+0.91}}$ | $82.20_{+0.99}$ |
| | Day 22 | $-88.57_{+58.85}$ | $-21.10_{+58.85}$ | $-0.08_{+0.04}$ | $-74.60_{+60.20}$ | $\mathbf{55.49_{+3.82}}$ | $52.88_{+9.75}$ | $55.47_{+8.34}$ | $80.43_{+1.26}$ |
| | Day 24 | $-39.52_{+86.25}$ | $-10.28_{+86.25}$ | $0.04_{+0.02}$ | $-14.52_{+76.57}$ | $62.52_{+2.39}$ | $70.13_{+2.49}$ | $\mathbf{71.32_{+1.75}}$ | $86.54_{+0.97}$ |
| | Day 25 | $-253.83_{+270.30}$ | $-64.67_{+270.30}$ | $0.10_{+0.03}$ | $-60.00_{+37.44}$ | $62.24_{+4.23}$ | $61.73_{+2.34}$ | $\mathbf{66.64_{+3.08}}$ | $85.80_{+1.08}$ |
| | Day 28 | $-107.64_{+124.47}$ | $-35.95_{+124.47}$ | $0.03_{+0.02}$ | $-46.10_{+74.64}$ | $48.82_{+18.62}$ | $66.01_{+2.87}$ | $\mathbf{71.38_{+2.27}}$ | $88.33_{+0.44}$ |
| | Day 29 | $-206.99_{+117.46}$ | $-64.32_{+117.46}$ | $-0.00_{+0.03}$ | $-42.48_{+106.10}$ | $61.51_{+1.61}$ | $60.82_{+1.80}$ | $\mathbf{64.21_{+1.06}}$ | $80.92_{+1.67}$ |
| | Day 31 | $-63.01_{+40.94}$ | $-81.41_{+40.94}$ | $0.04_{+0.05}$ | $-77.22_{+91.34}$ | $62.17_{+2.68}$ | $61.77_{+0.94}$ | $\mathbf{68.23_{+2.17}}$ | $81.64_{+1.08}$ |
| | Day 32 | $-417.39_{+295.63}$ | $-40.10_{+295.63}$ | $-0.06_{+0.04}$ | $-78.23_{+59.91}$ | $55.30_{+4.43}$ | $58.97_{+2.84}$ | $\mathbf{69.17_{+3.39}}$ | $83.36_{+1.36}$ |
| RT-M | Day 0 | $77.91_{+1.14}$ | $74.86_{+1.03}$ | $76.35_{+2.36}$ | $75.28_{+1.96}$ | $59.26_{+3.14}$ | $70.73_{+3.58}$ | $\mathbf{80.24_{+1.97}}$ | $79.69_{+2.68}$ |
| | Day 1 | $58.51_{+4.42}$ | $65.97_{+2.38}$ | $0.05_{+0.01}$ | $-130.22_{+20.98}$ | $57.83_{+3.07}$ | $66.04_{+3.67}$ | $\mathbf{69.54_{+2.55}}$ | $76.27_{+1.30}$ |
| | Day 38 | $-17.93_{+17.68}$ | $21.34_{+6.71}$ | $-0.31_{+0.11}$ | $-54.49_{+32.98}$ | $59.14_{+1.82}$ | $\mathbf{64.14_{+1.77}}$ | $61.91_{+1.04}$ | $68.68_{+1.97}$ |
| | Day 39 | $-104.81_{+99.91}$ | $-36.86_{+25.62}$ | $-0.14_{+0.12}$ | $-38.28_{+43.94}$ | $58.13_{+1.58}$ | $\mathbf{69.86_{+3.89}}$ | $68.78_{+1.68}$ | $78.03_{+1.28}$ |
| | Day 40 | $-14.81_{+47.15}$ | $2.63_{+20.16}$ | $0.06_{+0.04}$ | $-31.41_{+39.80}$ | $61.27_{+1.51}$ | $66.01_{+3.75}$ | $\mathbf{68.77_{+3.52}}$ | $83.55_{+1.52}$ |
| | Day 52 | $6.10_{+18.37}$ | $30.50_{+6.94}$ | $-0.16_{+0.05}$ | $-110.11_{+46.32}$ | $53.41_{+4.55}$ | $47.74_{+7.58}$ | $\mathbf{56.31_{+2.23}}$ | $61.36_{+3.15}$ |
| | Day 53 | $-47.05_{+63.72}$ | $42.33_{+4.84}$ | $-0.36_{+0.05}$ | $-112.86_{+31.80}$ | $53.65_{+2.69}$ | $61.96_{+6.85}$ | $\mathbf{68.49_{+1.15}}$ | $76.92_{+1.96}$ |
| | Day 67 | $-158.42_{+104.75}$ | $25.09_{+13.79}$ | $-0.30_{+0.07}$ | $-81.18_{+77.17}$ | $58.12_{+1.73}$ | $43.76_{+7.69}$ | $\mathbf{64.79_{+1.54}}$ | $74.83_{+0.97}$ |
| | Day 69 | $-101.08_{+32.58}$ | $-38.82_{+29.41}$ | $-0.22_{+0.04}$ | $-168.49_{+35.82}$ | $50.17_{+4.60}$ | $34.54_{+6.80}$ | $\mathbf{53.94_{+2.31}}$ | $66.07_{+2.27}$ |
| | Day 77 | $-280.39_{+104.05}$ | $-53.79_{+21.04}$ | $0.01_{+0.00}$ | $-63.76_{+54.85}$ | $52.39_{+2.32}$ | $33.38_{+7.72}$ | $\mathbf{56.90_{+2.28}}$ | $60.33_{+1.62}$ |
| | Day 79 | $-184.05_{+76.77}$ | $-47.01_{+13.77}$ | $-0.13_{+0.05}$ | $-46.66_{+49.72}$ | $55.01_{+3.05}$ | $36.83_{+12.53}$ | $\mathbf{58.35_{+2.40}}$ | $76.56_{+0.92}$ |

## 4.2 COMPARATIVE STUDY

**Baselines** We used the following methods as baselines, more details are shown in Appendix B.2:

**LSTM**(Hochreiter & Schmidhuber, 1997): We employed an unaligned LSTM as the decoder to evaluate the efficacy of alignment.

**CEBRA**(Schneider et al., 2023): CEBRA is a machine learning method that compresses time series to uncover hidden structures, demonstrating broad generalizability across various datasets and conditions.

**DAF**(Jin et al., 2022) DAF leverages an attention mechanism to extract domain-invariant features while retaining domain-specific details through a shared module, domain discriminator, and private modules.

**ERDiff**(Wang et al., 2023) ERDiff utilizes diffusion models to meticulously reconstruct spatio-temporal structures and seamlessly aligns them with the latent dynamics extracted from VAEs.

**NoMAD**(Karpowicz et al., 2022): NoMAD achieves signal alignment in neural manifolds by capturing the latent dynamics of neural population activities via LFADS (Pandarinath et al., 2018).

**Cycle-GAN** (Ma et al., 2023): This research employs Cycle-GAN to align the distributions of full-dimensional neural recordings at each time step.

**Results** We conducted quantitative comparisons using average $R^2$ of target domains. Day 0 corresponds to $\mathcal{D}_S$, while the other sessions represent $\mathcal{D}_T$. Results for CO-M and RT-M are presented in Table 1, with results for the CO-C dataset shown in Table 8. Our method consistently outperforms others across most sessions of the selected datasets. The unaligned LSTM and CEBRA designed for generalizable representations frequently fail over extended periods, demonstrating the need for distribution alignment to stabilize decoding performance. Regarding existing UDA-based alignment methods, HDA achieves over 6.00% and 10.00% higher $R^2$ on average compared to Cycle-GAN for CO-M and RT-M, respectively. Compared to NoMAD, we achieve over 10.00% and 8.00% higher on average. It can also be seen that HDA achieves much better performance than ERDiff and DAF. In addition, we visualized reach trajectories of CO-M integrated from the decoded cursor velocity. As shown in Appendix C.2, we observe that HDA yields more precise trajectories.

In addition, HDA consistently demonstrates effective performance across various source days. To validate this, we conducted additional experiments across all sessions, with the overall performance

illustrated in Fig. 6(a). We also conducted experiments using Day 0 as the source session, with source and target training ratios set at $0.1, 0.2, 0.4, 0.6$. As illustrated in Fig. 6(b), HDA shows effective performance even with a relatively small number of trials.

As for feature stability, we compared our MLE with those from ERDiff and NoMAD, as MLE can only be derived from sequential models. Fig. 7(d) shows the average MLE for each target session and their overall averages. A non-positive MLE typically indicates the stability of dynamical systems. We observed that HDA generally achieves the most stable latent space. Additionally, ERDiff exhibits greater instability compared to NoMAD, consistent with the $R^2$ score performance shown in Table 1. We further visualized latent trajectories of latent features $z(t)$ on Day22, Day24 from CO-M. Presented Fig. 3(c), we found that the divergent curves evolve over time to gradually approach each other. This indicates that the trajectories exhibit characteristics of Lyapunov stability.

**Computational Efficiency** We compared our efficiency with that of the baselines. Presented in Table 2, we found that HDA exhibits a similar parameter count to ERDiff and NoMAD, with greater time efficiency.

Table 2: List of computational efficiency with different methods

| Method | DAF | ERDiff | Cycle-GAN | NoMAD | **HDA** |
|---|---|---|---|---|---|
| Parameter Number (M) | 0.06 | 0.04 | 0.03 | 0.05 | **0.04** |
| Training Time per Epoch (s) | 0.15 | 0.28 | 0.02 | 3.77 | **0.06** |

## 4.3 ABLATION STUDY

We conducted an ablation study to confirm the effectiveness of HDA. Performance was evaluated based on the cross-session decoding and the stability of extracted dynamical latent features.

**Evaluation of Main Components** We specifically compared our full method against variations lacking raw neural signal adaptation (HDA-r), latent space decomposition (HDA-d), and semantic subspace adaptation (HDA-s). The cross-session decoding performance was validated on CO-C, CO-M, and RT-M datasets, with the results presented in Fig. 2(a). HDA performs the best, demonstrating the effectiveness of our main modules for stabilizing latent features. We observe that $R^2$ of HDA-r decreases the most, indicating that this step forms the foundation for better latent space alignment. Furthermore, HDA-d yields lower $R^2$, highlighting the advantages of latent space decomposition for more stable semantic features. Without the semantic subspace alignment, HDA-s performs second best on average, underscoring the necessity of further alignment within the decomposed subspace.

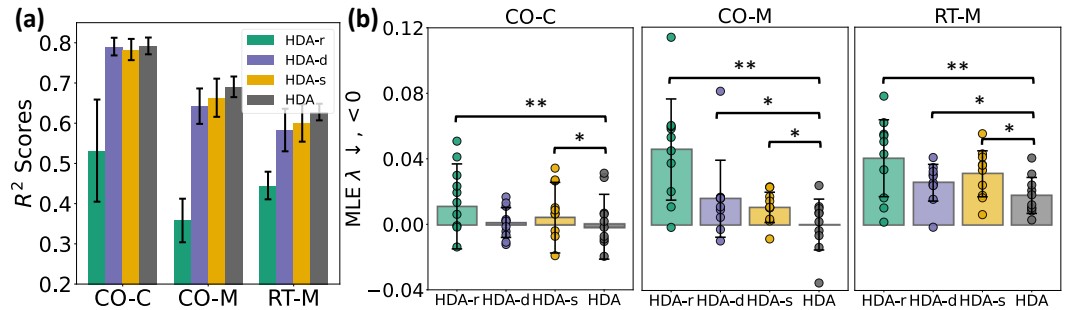

Figure 2: (a) $R^2$ scores for cross-session decoding, achieved by the variants HDA-r, HDA-d, HDA-s, and HDA across CO-C, CO-M, and RT-M datasets. (b) Comparison of the maximum Lyapunov exponent $\lambda$ with different methods on CO-C, CO-M, and RT-M datasets. Dots in various colors represent average MLE from an individual session. The symbols '*' and '**' denote significant $p$-values from paired $t$-tests, indicating $p < 5 \times 10^{-2}$ and $p < 1 \times 10^{-2}$, respectively.

**Evaluation of Each Loss Term** We further conducted a ablation study on each loss term. Specifically, we evaluated on $\mathcal{L}_y$, $\mathcal{L}_b$, and $\mathcal{L}_o$ with different weights $\lambda_y$, $\lambda_b$, and $\lambda_o$. The average $R^2$ scores (%) are listed below. As shown in Table 3, it can be seen that all these loss terms are necessary for optimizing the model performance.

**Stability Validation of Dynamical Latent Features** To validate the stability of dynamical latent features after our adaptation, we evaluated the dynamical system's stability based on the maximum Lyapunov exponent $\lambda$ as mentioned in Section 3.4. A non-positive MLE often indicates the stability of dynamical systems, achieving stable dynamical latent features (Wolf et al., 1985). More background information is shown in Appendix A.2. As depicted in Fig. 2(b), consistent with previous findings on the decoding stability, HDA-r emerges as the most unstable. This underscores the stabilizing effect of consistent input neural signals on latent features. Furthermore, semantic subspace alignment effectively enhances the dynamical stability of latent features, compared to HDA-d and HDA-s.

Furthermore, empirical results of pre-alignment are shown in Fig. 3(a) and (b). We found that $R^2$ and MLE demonstrate an upward trend with an increasing number of pre-alignment epochs. In addition, we also observed that the latent space alignment can enhance its dynamical stability. As depicted in Fig. 3(e), MLE converges to a non-positive value with an increasing number of alignment epochs. We conducted additional experiments on $R^2$ of test signals from the target session during latent space alignment as well. As depicted in Fig. 3(d), the curves on CO-M, and RT-M show successful convergences, indicating the training stability of HDA.

Table 3: Full ablation studies on different loss terms. Each weight is adjusted while keeping the other two fixed at 0.1.

| Data | $\lambda_o$ | | | | $\lambda_b$ | | | | $\lambda_y$ | | | |
|---|---|---|---|---|---|---|---|---|---|---|---|---|
| | 0 | 0.1 | 1 | 2 | 0 | 0.1 | 1 | 2 | 0 | 0.1 | 1 | 2 |
| CO-C | $79.24_{\pm1.90}$ | $79.60_{\pm1.68}$ | $\mathbf{79.92}_{\pm1.94}$ | $79.84_{\pm1.54}$ | $79.02_{\pm2.42}$ | $\mathbf{81.17}_{\pm1.98}$ | $79.53_{\pm3.04}$ | $78.99_{\pm3.12}$ | $-1.46_{\pm11.23}$ | $\mathbf{80.39}_{\pm2.20}$ | $79.53_{\pm2.90}$ | $78.57_{\pm3.00}$ |
| CO-M | $67.68_{\pm4.31}$ | $67.93_{\pm3.50}$ | $\mathbf{69.21}_{\pm2.89}$ | $68.38_{\pm4.24}$ | $66.75_{\pm4.25}$ | $69.49_{\pm3.81}$ | $69.10_{\pm3.92}$ | $\mathbf{69.60}_{\pm3.21}$ | $8.85_{\pm2.12}$ | $\mathbf{67.50}_{\pm4.35}$ | $67.38_{\pm4.14}$ | $66.36_{\pm4.86}$ |
| RT-M | $65.02_{\pm4.92}$ | $65.12_{\pm3.50}$ | $64.37_{\pm2.06}$ | $\mathbf{65.63}_{\pm5.29}$ | $60.06_{\pm2.07}$ | $\mathbf{63.86}_{\pm4.28}$ | $63.61_{\pm4.49}$ | $61.90_{\pm4.85}$ | $-1.02_{\pm0.72}$ | $\mathbf{62.16}_{\pm4.14}$ | $61.55_{\pm3.97}$ | $61.54_{\pm4.49}$ |

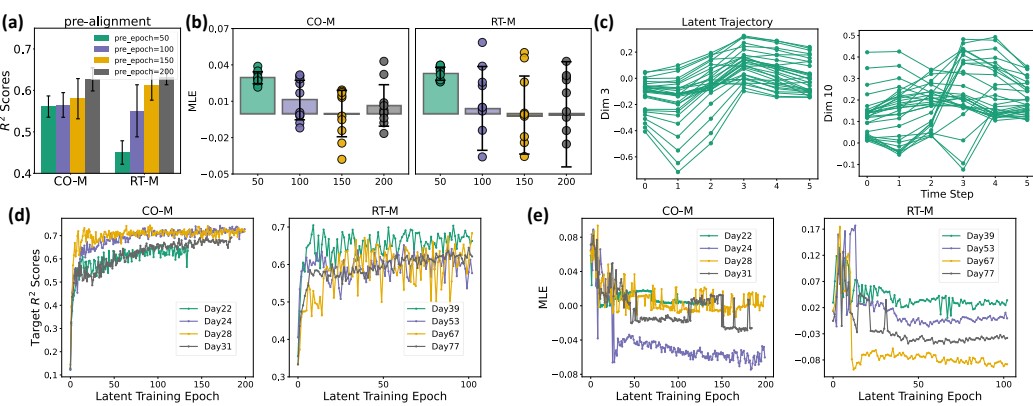

Figure 3: $R^2$ scores (a) and Maximum Lyapunov Exponent (MLE) (b) with varying pre-alignment training epochs (50, 100, 150, and 200) before the optimization. (c) Latent state trajectory visualizations ($z(t)$ from dimension 3 and 10) from CO-M on Day22 and 24. $R^2$ scores (d) and MLE (e) of test target trials during the latent space alignment on CO-M, and RT-M datasets, respectively.

## 5 CONCLUSIONS AND LIMITATIONS

In this study, we addressed the challenge of the decoding instability in BCIs caused by the variability of neural signals over time. We present a novel hierarchical domain adaptation (HDA) that focuses on neural dynamics. This framework utilizes causal architecture to extract dynamical latent features, and improves the feature stability based on the self-consistent alignment. The experimental analysis, supported by Lyapunov stability theory, demonstrate that our HDA can effectively improve the stability of dynamical systems, allowing for high-performance behavioral decoding for non-stationary neural signals. Our work successfully addressed the challenge of non-stationary neural signals, thereby potentially advancing the reliability of BCIs in chronic applications.

The limitations of this study are as follows. For scenarios with abundant data, HDA requires further verification to extract more generalizable features, enabling zero-shot inference. Additionally, the extension of HDA to other datasets involving humans needs validation.

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

# A    HIERARCHICAL DOMAIN ADAPTATION

## A.1    ARCHITECTURE DETAILS

We present the detailed architecture of our main modules as follows. The input neural signals have the shape of (Batch size=256, Window size=$w$, Number of channels=$L$). The latent dimensions of $\tilde{Z}_y$ and $\tilde{Z}_o$ are denoted as $k_y$, the dimension of latent states extracted by the nonlinear dynamical system as $k_h$. The dropout value is represented as $v_d$. The architectures of $E_\gamma$, $Q_\phi$, $R_\theta$, $D^y_{\beta_y}$, and $D^o_{\beta_o}$ can be seen in Table 4.

Table 4: Detailed Architectures of Modules

| | |
|---|---|
| $E_\gamma$ | LSTM($L$, $k_h$) |
| $Q_\phi$ | FC($k_h$, $2k_y$, $v_d$)×2 |
| $D^y_{\beta_y}$ | FC($k_y$, $k_y$, $v_d$), ReLU(), FC($k_y$, $k_y$, $v_d$), ReLU(), FC($k_y$, 1), Sigmoid() |
| $D^o_{\beta_o}$ | FC($k_y$, $k_y$, $v_d$), ReLU(), FC($k_y$, $k_y$, $v_d$), ReLU(), FC($k_y$, 1), Sigmoid() |

Here, we use the term FC to refer to fully connected layers, LSTM to represent Long Short-Term Memory layers, and ReLU and Sigmoid to denote the corresponding activation functions.

Moreover, default dimensions $k_y$, $k_h$, and value $v_d$ mentioned above are configured as shown in Table 5 according to different datasets.

Table 5: Default Value Setup on Different Datasets

| | $k_y$ | $k_h$ | $v_d$ |
|---|---|---|---|
| CO-C | 32 | 32 | 0.01 |
| CO-M | 32 | 32 | 0.01 |
| RT-M | 32 | 32 | 0.01 |

## A.2    LYAPUNOV STABILITY THEORY

### A.2.1    RELATED WORK

Lyapunov stability examines the stability of latent state trajectories within a dynamical system when its initial conditions or external inputs experience perturbations (Jiang & Wang, 2001). This concept, introduced by Lyapunov (Lyapunov, 1992), has been widely utilized in the stability analysis of various dynamical systems, including discrete linear systems (Goh, 1977) and nonlinear non-autonomous systems (Jiang et al., 1996). With the advent of deep learning, neural networks such as Recurrent Neural Networks (RNNs) and Long Short-Term Memory networks (LSTMs) are frequently employed to model complex nonlinear dynamical systems, with their hidden variables corresponding to the system's latent states. Some studies have integrated deep learning with Lyapunov stability to explore stability during network training (Engelken et al., 2023) and the network robustness (Ribeiro et al., 2020).

In the field of neuroscience, dynamical systems are frequently employed to model cognitive processes (Beer, 2000) and neural activities within the motor cortex (Ijspeert et al., 2013). The stability theory has also been leveraged to analyze these neural activities. For example, studies have identified that discrete attractors in the prefrontal cortex (Inagaki et al., 2019) are related to Lyapunov stability and lay the foundation for the working memory performance of animals undertaking delayed alternation tasks. Inspired by these insights and considering the presence of stable dynamical systems within the brain, we integrated the concept of Lyapunov stability with the process of extracting stable latent features from non-stationary neural signals.

### A.2.2 Rationale for Validation via Lyapunov Stability

In biological systems, similar behaviors often manifest analogous activities within neural populations. However, neural signals from the target domain may deviate from expected similarities with the source domain due to various factors. These stochastic factors can cause drifts in any stochastic dimensions. Here, we argue that Lyapunov stability effectively characterizes the stability of extracted latent features against random perturbations in original signals. That is to say, the enhancement in Lyapunov stability of the dynamical system indicates the stabilization of dynamical latent features. Therefore, we utilize Lyapunov stability as a tool for a quantitative representation of the system's stability to validate the stability of dynamical latent features.

### A.2.3 More explanations on maximum Lyapunov exponent (MLE)

Here, we give a theoretical explanation on how the pre-alignment of HDA improves dynamical stability. According to the definition in (Jiang & Wang, 2001), stability measures the distance between any two hidden states at time $t$, denoted as $z_i(t)$ and $z_j(t)$. Since these states are extracted using LSTMs, their distance can be expressed through the Lipschitz continuity of the activation layers:

$$\|z_i(t) - z_j(t)\| \leqslant \mathbf{K}_z + \mathbf{K}_i \mathbf{L}_a \|W_c\| \|x_i(t) - x_j(t)\|, \tag{6}$$

where $\mathbf{K}_z$ and $\mathbf{K}_x$ are constants independent of $\|x_i(t) - x_j(t)\|$, and $\mathbf{L}_a$ is the Lipschitz constant of activation functions. Thus, the pre-alignment, which helps minimize $\|x_i(t) - x_j(t)\|$, aids in controlling the upper bound of $\|z_i(t) - z_j(t)\|$, enhancing the efficiency of latent feature alignment.

The stability defined in (Jiang & Wang, 2001) can be determined based on (Wolf et al., 1985) to estimate the stability of $z(t)$ as follows:

**Step 1:**
Select $N$ sample points, denoted one as $z_1(t_0)$, find $j$ such that $j = \arg \min_k \|z_1(t_0) - z_k(t_0)\|$, and let $L_0(t_0) = \|z_1(t_0) - z_j(t_0)\|$.

**Step 2:**
Find $t_i$, for a given constant $\epsilon$, such that $t_0 \leqslant t < t_i$, $L_0(t) \leqslant \epsilon$; $L_0(t_i) > \epsilon$. Let $L'_0 = L_0(t_i)$. Continue with $z_1(t_i)$ as the next sample point following Step 1.

**Step 3:**
The maximum Lyapunov exponent(MLE) $\lambda$ is approximately as follows:

$$\lambda \approx \frac{1}{N \Delta t} \sum_{s=1}^{M} \log_2 \left( \frac{L'_0}{L_0(t_0)} \right),$$

where $\Delta t$ is the time step interval and $M$ is the number of steps in a single orbit.

## B Experimental Details

### B.1 Dataset Description

**CO-C&CO-M**. Monkeys C and M performed a center-out (CO) reaching task while grasping an upright handle. Monkey C used its right hand and Monkey M its left. Each trial began with the monkey moving its hand to the workspace center. Following a random wait, one of eight equally-spaced outer targets in a circular arrangement appeared. The monkey then held through a variable delay until an auditory go cue. To receive liquid reward, the monkey had to reach the outer target within 1.0 second and hold for 0.5 seconds.

**RT-M**. Monkey M also performed a random-target (RT) task, reaching to sequences of three targets presented in random screen locations to complete a trial. The RT task used the same apparatus as the CO reaching task. Each trial began with the monkey moving its hand to the workspace center. Three targets were then sequentially presented, and the monkey had to move the cursor into each within 2.0 seconds of viewing. As the target positions were randomized, the cursor trajectory took on a "random-target" form each trial.

**Detailed Preprocess Process**. For all datasets, we extracted trials from 'gocue time' to 'trial end' and preprocessed the neural signals by digitizing, bandpass filtering (250-5000 Hz), and spike detection

based on root-mean square activity thresholds. The data was then timestamped and smoothed with a Gaussian kernel to estimate firing rate over 50 ms bins.

## B.2 BASELINE IMPLEMENTATION

**CEBRA**. CEBRA is a sophisticated machine-learning technique developed for the analysis and compression of time series data, particularly enhancing the study of behavioral and neural data. This method is capable of uncovering hidden structures within data variability and has been successfully applied to decode activity in the mouse brain's visual cortex, even reconstructing what a subject has viewed. The code is available from https://github.com/AdaptiveMotorControlLab/cebra.

**DAF**. The Domain Adaptation Forecaster (DAF) utilizes abundant data from a relevant source domain to enhance performance in a target domain with limited data. DAF employs an attention-based shared module with a domain discriminator and private modules for each domain, promoting the extraction of domain-invariant latent features while simultaneously retraining domain-specific features. Our approach effectively aligns keys from the source and target domains, allowing for effective knowledge transfer despite differing characteristics. Extensive experiments show that DAF outperforms state-of-the-art methods on both synthetic and real-world datasets, and ablation studies confirm the efficacy of our design choices.

**ERDiff**. This work proposes leveraging diffusion models to first extract latent dynamic structures in the source domain, then recover them well in the target domain through maximum likelihood alignment. Empirical evaluation on synthetic and neural recording datasets demonstrates this approach outperforms others by better preserving latent dynamic structures longitudinally and between individuals. We implement this based on the openly available code at https://github.com/yulewang97/ERDiff.

**NoMAD**. NoMAD leverages the latent manifold structure inherent in neural population activity to establish a stable link between brain activity and motor behavior. It demonstrates the ability to provide accurate and highly stable behavioral decoding over extended periods, eliminating the need for supervised recalibration. In this study, we implemented NoMAD using the LFADS code available at https://github.com/arsedler9/lfads-torch/tree/main. As a result, there may be some deviations from the original implementation.

**Cycle-GAN**. This work proposes utilizing Cycle-GAN to align the distributions of full-dimensional neural recordings and stabilize the original decoding model without requiring recalibration. Through evaluating Cycle-GAN and a related approach (ADAN) on multiple monkey and task datasets, Cycle-GAN demonstrated superior performance for robustly maintaining BCI accuracy longitudinally without additional training. As the study utilizes the same datasets, we directly implement its openly available code from https://github.com/limblab/adversarial_BCI.

## B.3 TRAINING DETAILS

The main configurations for model training included the learning rate, weight decay parameters of the Adam optimizer, batch sizes, number of training epochs, and GPU hardware. Details of these hyperparameters are provided in Table 6.

Table 6: Detailed Training Setup

|      | Learning Rate | Weight Decay | Epoch Number | Batch Size | GPU |
|------|---------------|--------------|--------------|------------|-----|
| CO-C | 2e-3 | 1e-5 | 2500 | 256 | NVIDIA 3080Ti |
| CO-M | 2e-3 | 5e-7 | 2000 | 256 | NVIDIA 3080Ti |
| RT-M | 2e-3 | 5e-7 | 3000 | 256 | NVIDIA 3080Ti |

Main hyper-parameters, the signal window size ($w$), and the weights balancing terms in the final loss function ($\lambda_{y,b,o}$) are set as shown in Table 7.

## B.4 DETAILED TEST PROCEDURE

Specifically, during the test phase, we employed neural signals $\mathbf{X}^T$ from the target domain, which were not leveraged during the training phase, to evaluate the efficacy of our alignment

Table 7: Hyper-parameter Setup

|  | $w$ | $\lambda_y$ | $\lambda_b$ | $\lambda_o$ |
|---|---|---|---|---|
| CO-C | 6 | 1 | 1e-2 | 1 |
| CO-M | 6 | 1 | 1e-2 | 1 |
| RT-M | 5 | 1 | 1e-2 | 1 |

approach. This evaluation is based on the decoding performance, as represented by $\mathcal{L}_y$: $\mathcal{L}_y = \left\| \mathbf{Y}^T - C_\eta \left( Q_{\phi_y} \left( E_\gamma \left( G_\alpha \left( \mathbf{X}^T \right) \right) \right) \right) \right\|_2$, where $\mathbf{Y}^T$ signifies the actual reaching velocity corresponding to $\mathbf{X}^T$.

## C ADDITIONAL RESULTS

### C.1 COMPARATIVE STUDY

The comprehensive results of average $R^2$ scores for cross-session velocity decoding on the CO-C dataset are detailed in Table 8.

Table 8: Average $R^2$ Scores (%) of Cross-session Velocity Decoding

| Data | Session | LSTM | Cebra | DAF | ERDiff | NoMAD | Cycle-GAN | **HDA** | retrain |
|---|---|---|---|---|---|---|---|---|---|
| CO-C | Day 0 | $86.65_{+1.18}$ | $88.30_{+1.66}$ | $86.25_{+0.87}$ | $\mathbf{88.52}_{+0.72}$ | $31.99_{+9.45}$ | $78.29_{+1.93}$ | $86.66_{+0.29}$ | $86.66_{+0.29}$ |
| | Day 1 | $5.04_{+27.90}$ | $15.41_{+14.89}$ | $-6.40_{+4.97}$ | $-7.59_{+12.30}$ | $44.39_{+5.49}$ | $70.31_{+4.23}$ | $\mathbf{83.32}_{+0.77}$ | $86.03_{+0.56}$ |
| | Day 2 | $9.25_{+32.85}$ | $53.00_{+6.85}$ | $-5.86_{+3.90}$ | $6.03_{+8.44}$ | $31.53_{+6.13}$ | $80.82_{+1.36}$ | $\mathbf{84.84}_{+4.68}$ | $89.60_{+0.52}$ |
| | Day 3 | $-128.25_{+65.07}$ | $23.32_{+13.39}$ | $-2.09_{+2.34}$ | $6.32_{+13.51}$ | $25.11_{+12.50}$ | $68.66_{+2.24}$ | $\mathbf{77.69}_{+2.91}$ | $86.35_{+0.99}$ |
| | Day 9 | $-24.15_{+33.53}$ | $-5.20_{+21.77}$ | $-1.80_{+2.15}$ | $-76.27_{+50.66}$ | $38.72_{+6.35}$ | $74.84_{+1.52}$ | $\mathbf{84.14}_{+1.96}$ | $88.55_{+0.68}$ |
| | Day 10 | $-70.33_{+65.25}$ | $-2.22_{+20.13}$ | $-3.70_{+3.36}$ | $3.23_{+8.19}$ | $42.12_{+9.81}$ | $74.61_{+1.14}$ | $\mathbf{82.18}_{+1.17}$ | $89.19_{+0.80}$ |
| | Day 14 | $-65.46_{+24.55}$ | $-13.54_{+26.38}$ | $-0.87_{+0.82}$ | $-38.13_{+72.01}$ | $39.90_{+20.83}$ | $63.52_{+1.53}$ | $\mathbf{73.95}_{+2.67}$ | $85.16_{+0.64}$ |
| | Day 15 | $-32.08_{+24.64}$ | $-31.94_{+17.11}$ | $-4.45_{+2.39}$ | $-9.75_{+16.73}$ | $35.71_{+15.39}$ | $78.00_{+0.39}$ | $\mathbf{84.41}_{+0.68}$ | $91.39_{+0.56}$ |
| | Day 16 | $-123.74_{+63.89}$ | $-10.21_{+17.65}$ | $-2.26_{+1.04}$ | $-29.42_{+57.08}$ | $41.33_{+13.65}$ | $74.52_{+0.34}$ | $\mathbf{80.91}_{+0.95}$ | $90.80_{+0.34}$ |
| | Day 36 | $-70.67_{+99.37}$ | $-55.33_{+19.88}$ | $-4.24_{+3.58}$ | $-29.41_{+56.85}$ | $35.17_{+8.61}$ | $39.70_{+34.29}$ | $\mathbf{74.02}_{+2.78}$ | $89.56_{+0.51}$ |
| | Day 37 | $-29.54_{+59.36}$ | $-44.82_{+31.72}$ | $-3.78_{+3.13}$ | $-2.44_{+10.22}$ | $51.48_{+10.78}$ | $67.46_{+3.59}$ | $\mathbf{81.31}_{+1.67}$ | $91.80_{+0.42}$ |
| | Day 38 | $-112.02_{+132.39}$ | $-23.46_{+22.71}$ | $-2.53_{+1.87}$ | $-4.37_{+5.70}$ | $41.33_{+8.24}$ | $28.18_{+1.15}$ | $\mathbf{64.68}_{+2.72}$ | $77.45_{+0.48}$ |

### C.2 DECODED CURSOR VELOCITY VISUALIZATION

As shown in Fig. 4 and Fig. 5, we visualized reach trajectories of CO-M integrated from the decoded cursor velocity.

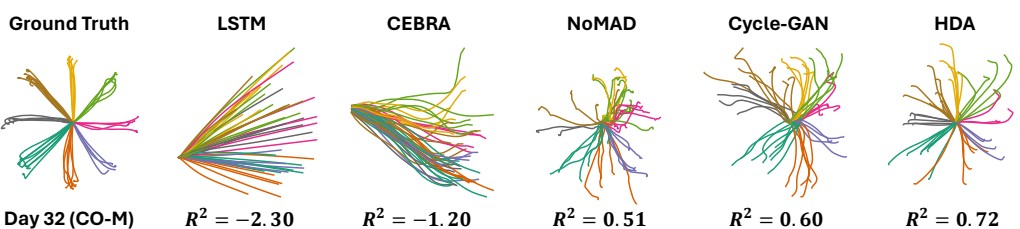

| Ground Truth | LSTM | CEBRA | NoMAD | Cycle-GAN | HDA |
|---|---|---|---|---|---|
| Day 32 (CO-M) | $R^2 = -2.30$ | $R^2 = -1.20$ | $R^2 = 0.51$ | $R^2 = 0.60$ | $R^2 = 0.72$ |

Figure 4: True and decoded cursor trajectories, integrated from the decoded velocity, are presented for baselines and HDA after aligning Day 32 to Day 0 on the CO-M. Different colors represent eight different reach directions.

### C.3 ADDITIONAL ANALYSIS ON HDA

We also conducted experiments using Day 0 as the source session, with source and target training ratios set at $0.1, 0.2, 0.4, 0.6$. As illustrated in Fig. 6(b), HDA shows effective performance even with a relatively small number of trials.

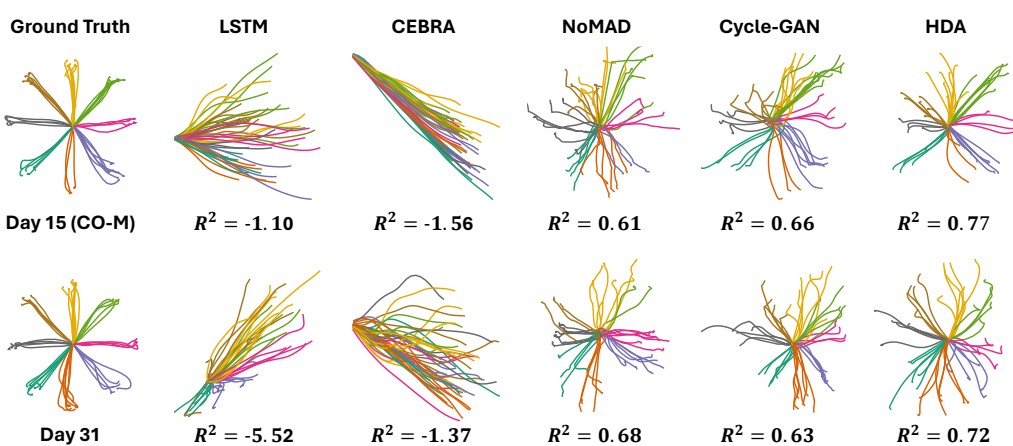

Figure 5: True and decoded cursor trajectories, integrated from the decoded velocity, are presented for baselines and HDA on Day 15 and Day31 of CO-M dataset. Different colors represent eight different reach directions.

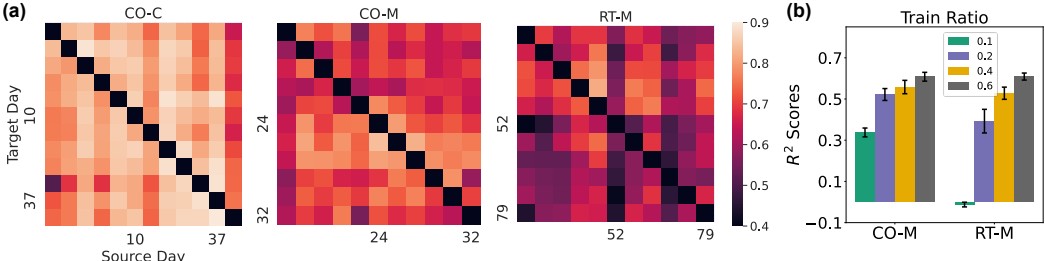

Figure 6: (a) Overall performance of average $R^2$ scores on all the sessions(days) from CO-C, CO-M, and RT-M. (b) HDA's performance on CO-M and RT-M at different training ratios (0.1, 0.2, 0.4, and 0.6). Here, the source session is Day0.

## C.4 FEATURE STABILITY COMPARISON

As for feature stability, we compared our MLE with those from ERDiff and NoMAD, as MLE can only be derived from sequential models. Fig. 7(d) shows the average MLE for each target session and their overall averages.

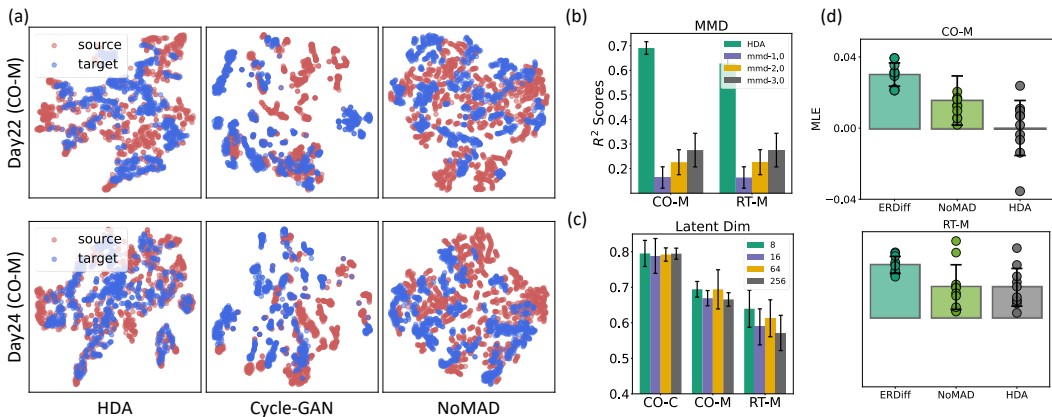

Figure 7: (a) t-SNE visualizations of HDA compared to Cycle-GAN and NoMAD on Day22 and 24 of CO-M. (b) $R^2$ scores of HDA and MMD with different Gaussian kernels ($\mu$=1.0, 2.0, 3.0). (c) $R^2$ score performance of HDA across different latent dimensions. ($\tilde{k}_o = \tilde{k}_y$ =8, 16, 64, 256). (d) Total average Maximum Lyapunov Exponent (MLE) for baselines containing sequential models (ERDiff, NoMAD) and HDA on CO-M and RT-M. Dots represent average MLE from an individual target session.

## C.5 LATENT FEATURE VISUALIZATION

As shown in Fig. 7(a), the t-SNE results are compared with the top two baselines, demonstrating our superior alignment performance.

## C.6 VISUALIZATION OF DYNAMICAL LATENT FEATURES

To examine our decomposition of the latent spaces, we selected CO-M as the representative dataset for visualization. We presented a visualization of the semantic dynamical latent features $\tilde{\mathbf{Z}}_y$, the domain-related latent features $\tilde{\mathbf{Z}}_o$, and original latent features $\mathbf{Z}$ from both the source and several target sessions, utilizing t-SNE for dimensionality reduction. These visualizations are depicted in Fig. 8 and Fig. 9. Our analysis reveals that the semantic latent features of the source and target sessions are closely aligned, while a discrepancy is observed in the distribution of the domain-related and original features. This observation suggests that HDA has effectively decomposed the latent space into semantic and domain-related subspaces.

## C.7 HYPER-PARAMETER SENSITIVITY ANALYSIS

The main hyper-parameters of our method include the signal window size ($w$), and the weights balancing terms in the final loss function ($\lambda_{b,o}$), and latent feature dimensions($\tilde{k}_{o,y}$). The results of their sensitivity analysis are shown in Tables 9 to 11, and Fig. 7(c).

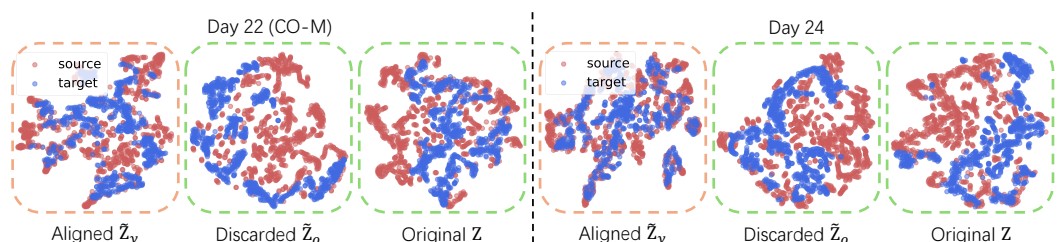

Figure 8: Visualizations via t-SNE are presented, depicting the semantic latent features $\tilde{\mathbf{Z}}_y$, the domain-related latent features $\tilde{\mathbf{Z}}_o$, and original latent features $\mathbf{Z}$. Each figure shows latent features from the source session and a specific target session from CO-M, represented by different colors.

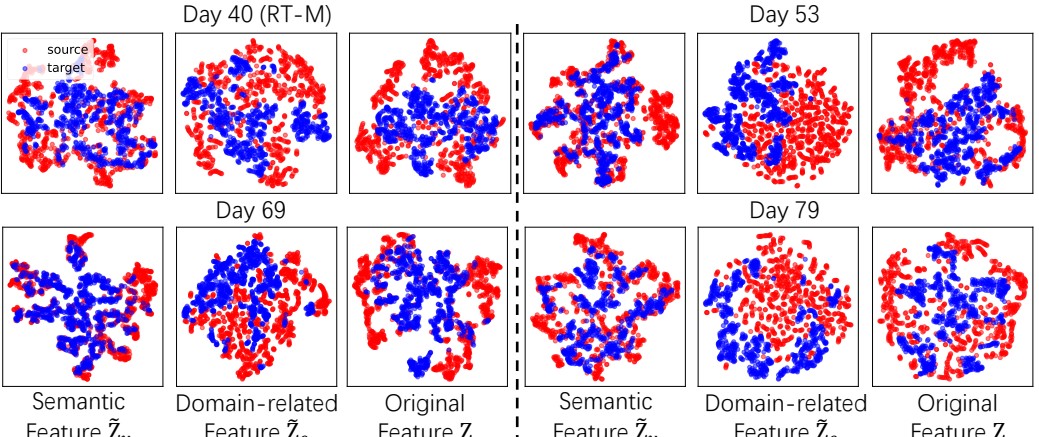

Figure 9: Visualizations via t-SNE are presented, depicting the semantic latent features $\tilde{\mathbf{Z}}_y$, the domain-related latent features $\tilde{\mathbf{Z}}_o$, and original latent features $\mathbf{Z}$. Each figure shows latent features from the source session and a specific target session from RT-M, represented by different colors.

Table 9: Average $R^2$ scores for different datasets with varying $\lambda_b$.

| $\lambda_b$ | CO-C | CO-M | RT-M |
|---|---|---|---|
| 0.0001 | $0.7912 \pm 0.0233$ | $0.6619 \pm 0.0502$ | $0.6448 \pm 0.0527$ |
| 0.001 | $0.7924 \pm 0.0237$ | $0.6641 \pm 0.0496$ | $0.6446 \pm 0.0536$ |
| 0.01 | $0.7984 \pm 0.0194$ | $0.6921 \pm 0.0289$ | $0.6437 \pm 0.0206$ |
| 0.1 | $0.8109 \pm 0.0177$ | $0.6838 \pm 0.0425$ | $0.6563 \pm 0.0529$ |

Table 10: Average $R^2$ scores for different datasets with varying $\lambda_o$.

| $\lambda_o$ | CO-C | CO-M | RT-M |
|---|---|---|---|
| 0 | $0.7924 \pm 0.0190$ | $0.6768 \pm 0.0431$ | $0.6502 \pm 0.0492$ |
| 0.1 | $0.7960 \pm 0.0168$ | $0.6793 \pm 0.0350$ | $0.6512 \pm 0.0350$ |
| 1 | $0.7992 \pm 0.0194$ | $0.6921 \pm 0.0289$ | $0.6437 \pm 0.0206$ |
| 2 | $0.7984 \pm 0.0154$ | $0.6838 \pm 0.0425$ | $0.6563 \pm 0.0529$ |

Table 11: Average $R^2$ scores for different datasets with varying $w$.

| $w$ | CO-C | CO-M | RT-M |
|---|---|---|---|
| 4 | $0.7640 \pm 0.0351$ | $0.6704 \pm 0.0339$ | $0.6273 \pm 0.0534$ |
| 5/6 | $0.7984 \pm 0.0194$ | $0.6921 \pm 0.0289$ | $0.6437 \pm 0.0206$ |
| 7 | $0.7769 \pm 0.0489$ | $0.6519 \pm 0.0781$ | $0.6559 \pm 0.0473$ |
| 8 | $0.8074 \pm 0.0348$ | $0.6703 \pm 0.0765$ | $0.6240 \pm 0.0618$ |

