# OpenReview forum: "Stabilized Neural Dynamics for Behavioral Decoding via Hierarchical Domain Adaptation"
_ICLR.cc/2025/Conference — Submitted to ICLR 2025_

### Official Review · Reviewer_NKh1 · 2024-10-19

**Soundness:** 2
**Presentation:** 3
**Contribution:** 2
**Rating:** 5
**Confidence:** 4

**Summary:**

Authors propose a neural alignment approach via hierarchical domain adaptation (HDA) in which they first align neural observations across Day 0 and Day K activity, followed by aligning their semantic latent activity. They show that HDA can perform alignment across various days of recordings better than baseline approaches.

**Strengths:**

Authors show that HDA can improve behavior decoding performance on Day K activity over baseline methods including single-session models of neural activity, e.g., CEBRA and vanilla LSTM, and other alignment approaches such as CycleGAN. Authors also provide a theoretical foundation based on Lyapunov theory for their 2-stage alignment approach.

**Weaknesses:**

- Major points:
  - It seems like the $E_\gamma$ is trained both on source and target domain data, rather than first training on Day 0 and aligning on Day K activity. However, I find this strategy counterintuitive for a neural alignment task. Shouldn’t authors first train their dynamic network $E_\gamma$ and decoder network $C_\eta$ on the source data initially? If the authors train their network every day from scratch with the new day’s data and source data, what is the advantage of the proposed framework over training completely a new model on new day’s data (as shown in Table 1 columns HDA and retrain)? The only advantage I can think of is the possibility of achieving better behavior decoding performance on Day K activity when the model is trained with extra data from Day 0, which does not seem to be the case. I understand that NoMAD also requires some information about Day 0 data to perform alignment such as latent factor distributions of Day 0 data, but such distributions can be computed across whole Day 0 data and stored to be used during alignment (which would not affect the alignment stage as latent factors for Day 0 data is obtained through frozen network during alignment). Even though this is not the case for ADAN, authors state that having a percentage of the Day 0 data for alignment (such as 50%, see https://github.com/limblab/adversarial_BCI/blob/main/ADAN_aligner.ipynb) is enough for the alignment stage after Day 0 training. I suspect the authors could have a similar strategy for their approach as they do not have any Day 0 training stage. I believe that this is particularly important since a BMI can be trained offline using a large dataset, but needing to have access to this large dataset during the alignment stage could limit the applicability of the decoder alignment. If the authors can, I think it is crucial to perform an ablation study on the amount of Day 0 data required for the alignment stage.
  - It seems like NoMAD performance does not change across different days, which contradicts the findings shown in the NoMAD paper. Beyond that, I am concerned about NoMAD’s performance on Day 0, which is basically LFADS as no alignment takes place for Day 0 and it has shown great promise across multiple neural datasets (see https://eval.ai/web/challenges/challenge-page/1256/leaderboard/3184). I am surprised that LFADS, i.e., NoMAD Day 0, is performing worse than vanilla **unidirectional** LSTMs. I appreciate that the authors implemented NoMAD themselves as the public repository is not available, but authors can compare against ADAN, whose repository is publicly available, and verify if ADAN will follow a performance pattern.
  - In line with my previous comment on LFADS, CEBRA performance on Day 0 also seems contradictory with prior findings as it again performs worse than vanilla LSTM. What positive/negative sampling strategy did the authors use while training CEBRA models?

- Minor points:
  - NDT and STNDT do not aim to build representations that are generalized (unlike claimed in L125-126) across sessions and animals, as they are trained for individual sessions.
  - Regarding the claim about ‘independence of decomposed subspaces’ for VAEs in L272-273, are the authors explicitly learning covariances between different dimensions of latent factors and pushing them towards zero with KL divergence, or treating each latent dimension separately as commonly done in VAEs? If latter, I suspect the model is essentially learning disentangled latent variables, rather, the model tries to push individual latent dimension variances to 1, and learned nothing explicitly about their covariances.

**Questions:**

- Is $\tilde{Z}_d^T$ in L286 a typo and correspond to $\tilde{Z}_O^T$?
- In L190-192, the authors mention aligning latent features spaces can be more challenging since spatiotemporal dependencies are more intricate, but for low dimensional latent factors (which is the case for authors as in L225), wouldn’t it be simple as they would capture less-noisy underlying patterns resulting in noisy spiking observations?
- Figure 6 is missing heatmap legends.

---

> ### Author Response · Authors · 2024-12-02
> **Response(1/2) to Reviewer NKh1**
>
> Thanks for your constructive comments and insightful suggestions. We have revised the manuscript accordingly, and provided a point-by-point response to your concerns.
>
> ### Major Weaknesses:
>
> 1. **Shouldn’t authors first train their dynamic network $E_\gamma$ and decoder network $C_\eta$ on the source data initially? If the authors train their network every day from scratch with the new day’s data and source data, what is the advantage of the proposed framework over training completely a new model on new day’s data (as shown in Table 1 columns HDA and retrain)?**
>
>    Thank you for the question. We employed adversarial alignment in both raw neural signal and latent spaces, training from scratch with unlabeled target data and labeled source data. This approach aligns with the traditional domain-adversarial training of neural networks (DANN) as detailed in Algorithm 1 on Page 10 of [1].
>
>    The advantage of training from scratch with the new day’s data and source data is that it does not require labels for the new day’s data. In contrast, the retraining in Table 1 is supervised. Our unsupervised alignment eliminates the need for labeled data, which is often scarce in neuroscience.
>
>    _[1] Ganin Y, Ustinova E, Ajakan H, et al. Domain-adversarial training of neural networks. Journal of machine learning research, 2016, 17(59): 1-35._
>
> 2. **I understand that NoMAD also requires some information about Day 0 data to perform alignment such as latent factor distributions of Day 0 data, but such distributions can be computed across whole Day 0 data and stored to be used during alignment (which would not affect the alignment stage as latent factors for Day 0 data is obtained through frozen network during alignment). Even though this is not the case for ADAN, authors state that having a percentage of the Day 0 data for alignment (such as 50%) is enough for the alignment stage after Day 0 training. I suspect the authors could have a similar strategy for their approach as they do not have any Day 0 training stage. I believe that this is particularly important since a BMI can be trained offline using a large dataset, but needing to have access to this large dataset during the alignment stage could limit the applicability of the decoder alignment. If the authors can, I think it is crucial to perform an ablation study on the amount of Day 0 data required for the alignment stage.**
>
>    Thanks for raising the unclear point, which we should be clearer. As mentioned in the response to the previous question, we trained from scratch using unlabeled target data and labeled source data, as the case for ADAN. Specifically, we utilized 80% of the trials (approximately 160) from a session as the source domain, which is feasible for storage. We agree that investigating the amount of Day 0 data required for the alignment stage is crucial and will address this in future work.
>
> 3. **It seems like NoMAD performance does not change across different days, which contradicts the findings shown in the NoMAD paper. Beyond that, I am concerned about NoMAD’s performance on Day 0, which is basically LFADS as no alignment takes place for Day 0 and it has shown great promise across multiple neural datasets (see https://eval.ai/web/challenges/challenge-page/1256/leaderboard/3184). I am surprised that LFADS, i.e., NoMAD Day 0, is performing worse than vanilla unidirectional LSTMs. I appreciate that the authors implemented NoMAD themselves as the public repository is not available, but authors can compare against ADAN, whose repository is publicly available, and verify if ADAN will follow a performance pattern.**
>
>    Thank you for pointing out the issue. We sincerely apologize for this oversight. We implemented NoMAD via publicly available codes of LFADS. Upon investigation, we found that the performance difference on Day 0 might stem from variations in hyperparameter configurations, as LFADS involves numerous hyperparameters. We will continue to optimize these configurations and include the corresponding results in future work.
>
>    For ADAN, given its similarity in adversarial alignment to Cycle-GAN and its inferior performance reported in the Cycle-GAN paper, we selected Cycle-GAN as the baseline.
>
> 4. **In line with my previous comment on LFADS, CEBRA performance on Day 0 also seems contradictory with prior findings as it again performs worse than vanilla LSTM. What positive/negative sampling strategy did the authors use while training CEBRA models?**
>
>    Thank you for raising this issue, and we sincerely apologize for the oversight. Upon re-running the publicly available codes for CEBRA, we found that the previous results were due to an overly small latent dimension. After adjusting the latent dimension to 32, CEBRA outperformed vanilla LSTMs on Day 0 of the CO-M dataset (79.24% vs. 74.18%) and achieved comparable results on the RT-M dataset (74.86% vs. 77.91%). These updated results have been incorporated into the current Table 1 on Page 8.

---

> ### Author Response · Authors · 2024-12-02
> **Response(2/2) to Reviewer NKh1**
>
> ### Minor Weaknesses:
>
> 1. **NDT and STNDT do not aim to build representations that are generalized (unlike claimed in L125-126) across sessions and animals, as they are trained for individual sessions.**
>
>    Thank you for the comment. We have revised the sentence as follows: _“In neural data analysis, researchers have focused on developing robust and generalizable representations using advanced architectures such as transformers (Ye & Pandarinath, 2021; Liu et al., 2022; Le & Shlizerman, 2022).”_ (Lines 118–121 on Page 3)
>
> 2. **Regarding the claim about ‘independence of decomposed subspaces’ for VAEs in L272-273, are the authors explicitly learning covariances between different dimensions of latent factors and pushing them towards zero with KL divergence, or treating each latent dimension separately as commonly done in VAEs? If latter, I suspect the model is essentially learning disentangled latent variables, rather, the model tries to push individual latent dimension variances to 1, and learned nothing explicitly about their covariances.**
>
>    Thank you for the insightful comment. In our work, we treated each latent dimension separately, as is commonly done in VAEs. This treatment of latent dimensions is also used in [1] (as presented in the TranSVAE_Video class of TranSVAE.py file from its public code [https://github.com/ldkong1205/TranSVAE/blob/main/exp/TranSVAE.py]) and [2] for representation disentanglement. We agree that it is important to determine whether the model itself is pushing the variances of individual latent dimensions to 1, and we plan to include more ablation studies to explore this in future work.
>
>    _[1] Wei P, Kong L, Qu X, et al. Unsupervised video domain adaptation for action recognition: A disentanglement perspective[J]. Advances in Neural Information Processing Systems, 2023, 36: 17623-17642._
>
>    _[2] Cai R, Li Z, Wei P, et al. Learning disentangled semantic representation for domain adaptation//IJCAI: proceedings of the conference. NIH Public Access, 2019, 2019: 2060._
>
> ### Questions:
>
> 1. **Is $\tilde{Z}_d^T$ in L286 a typo and correspond to $\tilde{Z}_o^T$?**
>
>    Thanks for raising the issue. We have corrected $\tilde{Z}_d^T$ to $\tilde{Z}_o^T$ in Line 294 on Page 6.
>
> 2. **In L190-192, the authors mention aligning latent features spaces can be more challenging since spatiotemporal dependencies are more intricate, but for low dimensional latent factors (which is the case for authors as in L225), wouldn’t it be simple as they would capture less-noisy underlying patterns resulting in noisy spiking observations?**
>
>    Thank you for the insightful comment. We believe that the less-noisy underlying patterns may not be easily captured using only the unlabeled target data. Additionally, we found that the underlying patterns extracted from pre-aligned spiking observations exhibited significantly greater stability across sessions, even without latent space alignment. This is validated by the results for HDA without raw space alignment (HDA-r) and HDA without latent space alignment (HDA-s), as shown in Figure 2(a) on Page 9.
>
> 3. **Figure 6 is missing heatmap legends.**
>
>    Thank you for pointing out the issue. We have added the heatmap legends, as shown in Figure 6(a) on Page 20.
>
> We sincerely appreciate your constructive comments and insightful suggestions, which are valuable for improving our work. Thanks for your time and consideration.

---

> > ### Comment · Reviewer_NKh1 · 2024-12-03
> >
> > I thank the authors for their response, however, hyperparameter optimizations for baseline models are pretty important and should not be considered as future work, since I believe that the baseline comparisons are very important for this submission, and NoMAD results still seem confusing. Overall, I will keep my score and in addition to providing stronger evidence on baselines, I recommend authors perform some ablation studies on the performance change with different amounts of Day 0 data used during alignment, and retraining models from scratch in an unsupervised setting, in addition to what they have done in Table 1, which can provide a more clear perspective on NoMAD's performance and HDA's actual performance gains.

---

> > > ### Author Response · Authors · 2024-12-03
> > >
> > > We sincerely appreciate your further feedback. Your comments and suggestions are valuable for improving our work.

---

### Official Review · Reviewer_oiYh · 2024-11-01

**Soundness:** 2
**Presentation:** 3
**Contribution:** 2
**Rating:** 5
**Confidence:** 3

**Summary:**

The paper created a hierarchical architecture for the alignment of the dynamical latent features. GAN is used for for the raw signal to match the distribution of the the source signal, semantic latent has been decomposed. The representation has been disentangled to get the domain-invariant features. The authors did experiments to show the decoding accuracy and ablation study on the each loss terms.

**Strengths:**

The paper uses simple networks to retrieve the neural representations and latents and fair experiments to compare with the existing models. The experiment results are clearly explained. The formulas are well notations. The frame of the paper is well presented.

**Weaknesses:**

The paper seems combining several things together, but kind of redundant work. First, GAN is used for matching the distribution, then a VAE is used for retrieving the latent. There is no need to use GAN since GAN and VAE do the same thing to approach the distribution, and for the latent decomposition in the same level, that does not change anything. The whole algorithm training process, need to alternately train several networks which seems not very efficient.

**Questions:**

One page 9, in the part Evaluation of Each Loss Term, the authors says from Table 3, all the loss terms are necessary. For the coefficient of the three loss terms, should they be given together as a tuple for one R square value? But in the table, I wander how the R square value is obtained giving only one parameter, but the other two are not given. Besides, to make a fire comparison, for example, setting the sum of the three coefficient to be one, and either fix the values or give a schedule for that to obtain the weights or necessaries? Also, from Table 7, $\lambda_b$ is relative small that could be omitted compared with the other two, which is a contradiction with the previous claim.

For the distribution divergence, can you give a reason why you choose the chi-square instead of KL divergence?

For the Lyapunov stability, since you are using the LSTM network, can you generalize it for complex network for rich representation?

---

> ### Author Response · Authors · 2024-12-02
> **Response to Reviewer oiYh**
>
> Thanks for your constructive comments and insightful suggestions. We have gone through the review, and provided a point-by-point response to your concerns.
>
> ### Weaknesses:
>
> 1. **The paper seems combining several things together, but kind of redundant work. First, GAN is used for matching the distribution, then a VAE is used for retrieving the latent. There is no need to use GAN since GAN and VAE do the same thing to approach the distribution, and for the latent decomposition in the same level, that does not change anything.**
>
>    Thanks for the insightful comment. Despite the VAE-imposed constraint requiring latent variables to follow a Gaussian distribution, we observed that the decomposed latent variables were not fully aligned under this constraint. This is evidenced by the drop in cross-session performance when HDA is used without latent space alignment via GANs, as shown in Figure 2(a) (HDA-s), particularly for the CO-M and RT-M datasets.
>
>    The latent space decomposition was performed to achieve representation disentanglement, extracting latent variables directly related to behavioral variables. This step helps remove the encoding of task-irrelevant perceptual information and environmental feedback within the latent dynamics. Its effectiveness is demonstrated by the performance of HDA-d in Figure 2(a).
>
> 2. **The whole algorithm training process, need to alternately train several networks which seems not very efficient.**
>
>    Thank you for the comment. Although training was performed alternately, the efficiency remains acceptable since the alignment relies on MLPs with relatively few parameters. As shown in Table 2 (Page 9), HDA achieved shorter training times per epoch on the same device compared to methods based on diffusion models (ERDiff) and transformers (DAF).
>
> ### Questions:
>
> 1. **One page 9, in the part Evaluation of Each Loss Term, the authors says from Table 3, all the loss terms are necessary. For the coefficient of the three loss terms, should they be given together as a tuple for one R square value? But in the table, I wander how the R square value is obtained giving only one parameter, but the other two are not given.**
>
>    Thank you for pointing out this unclear point. The coefficient of each individual loss term was adjusted while the other two were kept fixed at 0.01. We have added this information to the legend of Table 3 on Page 10.
>
> 2. **Besides, to make a fire comparison, for example, setting the sum of the three coefficient to be one, and either fix the values or give a schedule for that to obtain the weights or necessaries?**
>
>    Thank you for pointing out this issue. We agree that setting the sum of the three coefficients to be one provides more fair comparisons. Relevant ablation studies with the sum of the three coefficients to be one will be provided in the future work.
>
> 3. **Also, from Table 7, $\lambda_b$ is relatively small that could be omitted compared with the other two, which is a contradiction with the previous claim.**
>
>    Thank you for pointing out this issue. We sincerely apologize for the incorrect values in Table 7. The correct values are $\lambda_b$=0.1, $\lambda_o$=0.01 and $\lambda_y$=0.01, as these yield relatively better results, as shown in Table 3.
>
> 4. **For the distribution divergence, can you give a reason why you choose the chi-square instead of KL divergence?**
>
>    Thank you for raising these unclear points. The Chi-squared divergence is a specific case of $f$-divergence. We chose LSGANs based on the Chi-squared divergence, rather than naïve GANs based on KL-divergence, to ensure greater gradient stability during alignment. The relevant explanations have been added in Lines 200–205 on Page 4, as shown below:
>
>    _"However, since $f$-divergence is difficult to compute directly, we employed GANs to implement alignment based on $f$-divergences in an indirect manner. Given that naive GANs often suffer from training instability, we used LSGANs (Mao et al., 2017) based on the $\chi^2$ divergence, which is a specific case of $f$-divergence. The benefits of alignment based on $f$-divergences are demonstrated in Figure 7(b)."_
>
> 5. **For the Lyapunov stability, since you are using the LSTM network, can you generalize it for complex network for rich representation?**
>
>    Thank you for the question. The Lyapunov stability is defined for dynamical systems, which do not necessarily need to be modeled by LSTM networks. Thus, we believe it can be generalized to more complex networks capable of constructing dynamical systems, such as RNNs and diffusion models. Further empirical validation using these more complex networks will be provided in future work.
>
> Thanks a lot for your constructive comments and insightful suggestions, which are valuable for improving our work. Thanks for your time and consideration.

---

### Official Review · Reviewer_GCum · 2024-11-03

**Soundness:** 3
**Presentation:** 2
**Contribution:** 3
**Rating:** 5
**Confidence:** 4

**Summary:**

The authors develop a model that stabilizes neural dynamics across recording sessions from multiple days. The authors first explain why this is an issue in real-life scenarios, and then develop a model that significantly improves decoding performance when transferred from one day to new days.

**Strengths:**

I think the authors do a good job explaining what types of methods exist to tackle this problem, from models that perform alignment on the neural manifold to methods that align data in the original data space. Moreover, the authors compare their model against good baselines, and the results are extremely encouraging. I have some comments about the results, but if those are addressed the contribution can be very big, given the importance of the problem the authors try to solve in this paper. I think the paper is relatively original, but will be helped if the authors improve its clarity.

**Weaknesses:**

Major weakness(es): \
I believe there are a few issues that together form a large reason why I would not currently recommend acceptance of this paper. First, the authors should improve the structure of the paper, while reading it I felt like the authors jumped back and forth between topics in the paper, e.g, the introduction and related work have a lot of overlap, but read disjointed. Additionally, I think the authors do not explain their intuition behind certain model choices enough, it may not be clear to readers why the authors are decomposing their dynamical latent space for example. Similarly, the theoretical verification of dynamical feature stability does not seem to be one of the main contributions of the paper, so I would move this section to the Appendix and instead focus more on how the authors are measuring stability, which is new and original for papers that try to stabilize neural dynamics across sessions. In terms of writing clarity I would lastly recommend that the authors improve their mathematical notation, the authors often do not use colloquial names but math symbols when referring to components in their model which is confusing, and the math symbols themselves are sometimes also confusing, e.g. the authors use $\mathbf{Z}, \mathbf{\tilde{Z}}, \mathbf{\tilde{Z}}_d^T, \mathbf{Z^S}, \mathbf{\tilde{Z}}_y^S $, and more all in a short section of text (L277-312). Additionally, the authors use $ \cdot^T $ as notation (e.g. $ X^T$, which is confusing because the T is normally used as a transpose operator. I think this paper could be improved tremendously by moving much of the math to the appendix (also because some of the equations are adapted from previous work, e.g. equations 1, 3, 4 are adapted from work the authors cite), and focusing on motivations for components, see [1] Section 3.3 for some tips on improvements. An intuitive explanation of many of the model’s components will greatly improve the readability of the paper and although it is good to adapt math from previous work and use it in your paper, in its current form the notation/equations reduce the clarity of the work.

Minor weaknesses:
1. The authors mention that their model is causal and thus better for real-time BCI applications. However, their model requires training data from a future day to be applicable to that day, so I don’t see how this would lead to real-time application of the current model. Specifically, the authors mention on Lines 380-382 that they use 20% of the sessions of the day that they want to transfer to as a test set. This means that without first training on the 80% of the unlabeled data on that day, the model cannot do any decoding yet. This in itself is not an issue, but one of the claims of the paper in the introduction is that “… we have the potential to meet the real-time operational requirements of BCIs”, which is a claim I would weaken given that atleast at the start of the day, the model requires training data to align first.
2. The authors do not mention or compare to relevant work in neural population decoding [2]
3. L271-272, the authors should cite disentanglement works, e.g. [3, 4, 5] for the independence claim.
4. The authors should explain their intuition for using Chi-squared divergence more, not every reader may be familiar with Chi-squared divergence. Additionally, the authors should explain why they believe it is more accurate in measuring distribution discrepancies, since although Figure 7B (as mentioned on Line 199) does show that the author’s alignment method performs better, it is not immediately clear why this is the case and how other divergence methods compare to Chi-squared divergence.
5. Instead of using training time the authors should also (and more importantly) report the number of FLOPs since training time is very dependent on differences in hardware.
6. The authors do not compare their results with other methods for Figure 6.


Spelling/grammar:
- L144/145: “… over time as an UDA…” -> a UDA
- L175-177: “The aligned neural signals are then serve as...” -> are then provided
- L192: “This identification may help for aligning…” -> may help to align
- L245-246: “… can provide the dynamical system with an more efficient…” -> a more efficient

[1] Lipton, Z. C., & Steinhardt, J. (2019). Troubling Trends in Machine Learning Scholarship: Some ML papers suffer from flaws that could mislead the public and stymie future research. Queue, 17(1), 45-77. \
[2] Azabou, M., Arora, V., Ganesh, V., Mao, X., Nachimuthu, S., Mendelson, M., ... & Dyer, E. (2024). A unified, scalable framework for neural population decoding. Advances in Neural Information Processing Systems, 36. \
[3] Higgins, I., Matthey, L., Pal, A., Burgess, C. P., Glorot, X., Botvinick, M. M., ... & Lerchner, A. (2017). beta-vae: Learning basic visual concepts with a constrained variational framework. ICLR (Poster), 3. \
[4] Burgess, C. P., Higgins, I., Pal, A., Matthey, L., Watters, N., Desjardins, G., & Lerchner, A. (2018). Understanding disentangling in $\beta $-VAE. arXiv preprint arXiv:1804.03599. \
[5] Higgins, I., Amos, D., Pfau, D., Racaniere, S., Matthey, L., Rezende, D., & Lerchner, A. (2018). Towards a definition of disentangled representations. arXiv preprint arXiv:1812.02230.

**Questions:**

1. On line 348-349 the authors claim that pre-alignment helps minimize $ || x_i(t) - x_j(t)|| $, I am just wondering why that’s true. It seems to me that pre-alignment helps make the data more similar to the original training data, but there is no guarantee that in the original training data $ ||x_i(t) - x_j(t)|| $ is smaller than in the data for a new day.
2. The authors perform alignment between the label subspace from day 0 to a new day (see Figure 1). In that case: do the authors assume that the same activities are performed in the same order on day 0 and a new day? This seems to be a limiting assumption if that is the case. Comparatively, from reading NoMad, the authors assume that the distribution of all activities on one day are similar to all activities on another day, which is a less restrictive assumption. Can you clarify what assumption is accurate for your model?
3. The authors hypothesize in section 3.2.3 that drifts of dynamical latent features in the target domain primarily stem from latent variables that are loosely connected to observable behavioral variables. Can the authors expand on how they test this hypothesis.

---

> ### Author Response · Authors · 2024-12-02
> **Response(1/3) to Reviewer GCum**
>
> Thanks for your constructive comments and insightful suggestions. We have made revisions accordingly in the current manuscript, and provided a point-by-point response to your concerns.
>
> ### Major Weaknesses:
>
> 1. **The introduction and related work have a lot of overlap, but read disjointed.**
>
>    Thank you for the comment. We have removed the overlapping content in the Related Work and now focus solely on introducing works related to latent neural dynamics (starting from Line 94 on Page 2), which were not elaborated on in the Introduction.
>
> 2. **Additionally, I think the authors do not explain their intuition behind certain model choices enough, it may not be clear to readers why the authors are decomposing their dynamical latent space for example.**
>
>    Thank you for pointing out this issue. Our intuition behind decomposing dynamical latent spaces is as follows: _“When performing a specific task, the brain processes a wide range of information, including perception, decision-making, environmental cues, feedback, and more. For instance, task-irrelevant perceptual information and environmental feedback are also encoded within the latent dynamics. By decomposing these latent spaces to remove irrelevant components, we aim to reduce variability within the latent dynamics, thereby improving alignment of latent spaces.”_ These explanations have been added to Lines 242–246 on Page 5. Additional intuitions underlying certain model choices will be incorporated into the manuscript in future work. Thanks a lot.
>
> 3. **Similarly, the theoretical verification of dynamical feature stability does not seem to be one of the main contributions of the paper, so I would move this section to the Appendix and instead focus more on how the authors are measuring stability, which is new and original for papers that try to stabilize neural dynamics across sessions.**
>
>    Thank you for the suggestion. We have moved the theoretical verification of dynamical features to Appendix A.2.3 and expanded the discussion on how to measure stability in the current Section 3.4 (Lines 354-360 on Page 7). The updated section now reads as follows: _“Furthermore, the stability defined above can be determined using a Lyapunov function $ V(z) $. Given an equilibrium point $ z^* $ of the system, the following equations are satisfied: **(1)** $ V(z^*) = 0 $, **(2)** $ \dot{V}(z^*) = 0 $, **(3)** $ V(z) > 0 $ for all $ z \neq z^* $, **(4)** $ \dot{V}(z) < 0 $ for all $ z \neq z^* $. It is known that $ V(z) = \frac{1}{2} z^T z $ is one of the functions that meet the conditions. However, directly calculating complex $ V(z) $ can be difficult. Therefore, we used the method based on (Wolf et al., 1985) to estimate the stability of $ z(t) $ using the maximum Lyapunov exponent (MLE).”_
>
> 4. **In terms of writing clarity I would lastly recommend that the authors improve their mathematical notation, the authors often do not use colloquial names but math symbols when referring to components in their model which is confusing, and the math symbols themselves are sometimes also confusing.**
>
>    Thanks for raising the unclear points, which we should be clearer. We will correct the mathematical notation to eliminate these confusions in future work.
>
> 5. **I think this paper could be improved tremendously by moving much of the math to the appendix (also because some of the equations are adapted from previous work).**
>
>    Thank you for the insightful suggestion. We sincerely apologize for the excessive "mathiness". We will reorganize the equations and focus more on the motivations behind main components to improve the clarity of this work.

---

> ### Author Response · Authors · 2024-12-02
> **Response(2/3) to Reviewer GCum**
>
> ### Minor weaknesses:
>
> 1. **The authors mention that their model is causal and thus better for real-time BCI applications. However, their model requires training data from a future day to be applicable to that day, so I don’t see how this would lead to real-time application of the current model.**
>
>    Thank you for raising these unclear points. The real-time application discussed in our work is based on short-time causal windows in individual trials. The short-time windows are different from most existing methods like ERDiff, which typically require the entire trial for decoding after alignment. Specifically, our HDA approach can directly perform decoding via short-time causal windows without waiting for the full trials after alignment.
>
> 2. **The authors do not mention or compare to relevant work in neural population decoding [2].**
>
>    Thank you for raising this issue. We have added the relevant reference in Lines 120–122 on Page 3.
>
> 3. **L271-272, the authors should cite disentanglement works, e.g. [3, 4, 5] for the independence claim.**
>
>    Thank you for pointing out the issue. The disentanglement works have been added in Lines 269–270 on Page 5 for the independence claim.
>
> 4. **The authors should explain their intuition for using Chi-squared divergence more, not every reader may be familiar with Chi-squared divergence. Additionally, the authors should explain why they believe it is more accurate in measuring distribution discrepancies, since although Figure 7B (as mentioned on Line 199) does show that the author’s alignment method performs better, it is not immediately clear why this is the case and how other divergence methods compare to Chi-squared divergence.**
>
>    Thank you for pointing out this unclear point. We have expanded our intuition for using Chi-squared divergence as follows: Other divergence methods, such as MMD, often rely on sufficient statistics like the mean, which characterize the collective properties of random variables. These methods tend to be less sensitive to infrequently occurring values,  which are common and crucial in biological systems. Therefore, we use $f$-divergence, which is based on probability density functions of individual samples, to measure the discrepancy between distributions.
>
>    Moreover, since $f$-divergence is challenging to compute directly, we employed GANs to implement alignment based on $f$-divergences in an indirect manner. To ensure greater stability during training, we utilized LSGANs instead of naïve GANs for alignment via $\chi^2$ divergence, a specific case of $f$-divergence. The relevant explanations have been added in Lines 198–205 on Page 4, as shown below:
>
>    _"Therefore, we chose the $f$-divergence, which is based on probability density functions, to measure the discrepancy between distributions. However, since $f$-divergence is difficult to compute directly, we employed GANs to implement alignment based on $f$-divergences in an indirect manner. Given that naive GANs often suffer from training instability, we used LSGANs (Mao et al., 2017) based on the $\chi^2$ divergence, which is a specific case of $f$-divergence. The benefits of alignment based on $f$-divergences are demonstrated in Figure 7(b)."_
>
> 5. **Instead of using training time the authors should also (and more importantly) report the number of FLOPs since training time is very dependent on differences in hardware.**
>
>    Thank you for the comment. Since we ran HDA and the baselines on the same devices, we only reported the training time per epoch. Additional results on FLOPs will be included in future revisions.
>
> 6. **The authors do not compare their results with other methods for Figure 6.**
>
>    Thank you for the suggestion. We will include a comparison with the baselines for Figure 6 in future work.

---

> ### Author Response · Authors · 2024-12-02
> **Response(3/3) to Reviewer GCum**
>
> ### Questions:
>
> 1. **On line 348-349 the authors claim that pre-alignment helps minimize $\||x_i (t)-x_j (t)\||$, I am just wondering why that’s true. It seems to me that pre-alignment helps make the data more similar to the original training data, but there is no guarantee that in the original training data $\||x_i (t)-x_j (t)\||$ is smaller than in the data for a new day.**
>
>    Thank you for raising this issue. The reason lies in the fact that pre-alignment is based on the Chi-squared divergence, whose values are always lower-bounded by the corresponding total variation, i.e., $\||x_i (t)-x_j (t)\||$. Therefore, minimizing the Chi-squared divergence during pre-alignment can be approximately interpreted as minimizing $\||x_i (t)-x_j (t)\||$. The lower-bound relationship is introduced in Eq. (7.26) of Proposition 7.2 from the lecture notes [https://people.lids.mit.edu/yp/homepage/data/LN_fdiv.pdf].
>
> 2. **The authors perform alignment between the label subspace from day 0 to a new day (see Figure 1). In that case: do the authors assume that the same activities are performed in the same order on day 0 and a new day? This seems to be a limiting assumption if that is the case. Comparatively, from reading NoMad, the authors assume that the distribution of all activities on one day are similar to all activities on another day, which is a less restrictive assumption. Can you clarify what assumption is accurate for your model?**
>
>    Thank you for the insightful question. Since we divided each trial into short time windows, the corresponding label $y_i$ is a single value representing the 2D velocity of cursors and does not depend on the order of activities. As a result, the assumption of same behavior tasks orders across days is not required for our alignment.
>
>    Given that low-dimensional manifolds are shared among similar activities, as noted by NoMAD, we assume that the dynamical semantic space associated with similar activities is also shared. Consequently, the latent variables capable of decoding the individual 2D velocities in center-out trials can also decode similar behavioral variables, such as 2D positions in random-target trials. We agree that further empirical studies are necessary to validate the applicability of this assumption.
>
> 3. **The authors hypothesize in section 3.2.3 that drifts of dynamical latent features in the target domain primarily stem from latent variables that are loosely connected to observable behavioral variables. Can the authors expand on how they test this hypothesis.**
>
>     Thank you for the question. We found it challenging to directly test this hypothesis. However, the performance decrease observed in HDA without latent space decomposition (shown as HDA-d in Figure 2(a)) demonstrates the effectiveness of latent variables directly related to behavioral labels, indirectly supporting the hypothesis. We will provide more direct empirical validation in future work.
>
> Thanks once again for your constructive comments and insightful suggestions, which are valuable for improving our work. Thanks for your time and consideration.

---

> ### Comment · Reviewer_GCum · 2024-12-03
> **Response to rebuttal**
>
> I want to thank the authors for responding to my review. I think in many cases, especially in terms of writing, the authors have been able to improve their paper. One comment I would make about the new writing is that I think the authors should refrain from claiming that they "... propose a novel way to measure feature stability... ", although it is (in my reading of the literature) new specifically for BCI evaluation, it is in itself not necessarily a new measure.
>
> However, the authors leave many of the changes in response to my main weaknesses as future work. Moreover, the response to Q2 makes me less confident in the proposed method, since it implies that the authors use the labels for each short time window to align data between days. In a real-life scenario (exactly where a method like the proposed method would be most relevant), we do not have access to labels since we do not exactly know the movements of a person/animal. The authors should clear this up in further revisions of their work. I have therefore decided not to increase my score.
>
> To respond to a few of the other points the authors make:
> - MW1: Although I understand the authors use short time windows and thus do not require the full timeseries to infer predicted labels, they do currently use 80% of the unlabeled session to align the data of a certain day to the training data. In a real-life scenario, this means that 80% of a session needs to be recorded before the BCI 'works', hence making it non-real-time. In a future revision, the authors can potentially test how sensitive their method is to the % of unlabeled data from a future session.
> - MW4: I understand why the authors have used an f-divergence, however, there are many more f-divergences than Chi-squared divergence (e.g. KL-divergence, total variation etc.), can the authors argue why they pick Chi-squared divergence? The authors could also compare to other f-divergences to strengthen their argument further.
> - Q1: The link the authors provide results in a 404 not found. Can the authors re-link the source and cite it in the paper. I am not sure I understand the authors' explanation, if the Chi-squared divergence lower bounds the total variation, how do we get to a minimization of the upper bound for $||x_i(t) - x_j(t)||$? If the lower bound of $||x_i(t) - x_j(t)||$ is reduced, its maximum does not necessarily reduce as well.

---

> > ### Author Response · Authors · 2024-12-03
> >
> > We sincerely appreciate your further feedback. Your comments and suggestions are valuable for improving our work.
> >
> > Regarding MW1, we have included results on the percentage of unlabeled data from a future session. As shown in Figure 6(b), we found that HDA's performance remains relatively stable above 20% (approximately 40 trials).
> >
> > For Q1, we apologize for the 404 error. The correct link is provided below:
> > https://people.lids.mit.edu/yp/homepage/data/LN_fdiv.pdf. We will include this source in our updated manuscript. The minimization of the upper bound for $||x_i(t)-x_j(t)||$ ensures a relatively small upper bound, which helps alleviate significant deviations in the original spaces. Reducing the lower bound of $||x_i(t)-x_j(t)||$ may not effectively prevent these deviations.

---

### Official Review · Reviewer_BT8N · 2024-11-04

**Soundness:** 3
**Presentation:** 2
**Contribution:** 2
**Rating:** 5
**Confidence:** 4

**Summary:**

Summary: In this paper, the authors propose a method named HDA for the domain adaptation of high-dimensional neural observation data by performing the alignment in both the raw neural space and the latent space. It also links the connection to the Lyapunov exponent theory. In the emprical evalution part, the decoding results of HDA outperforms other baseline methods and manifest robustness.

**Strengths:**

1. How to resolve the domain turnover issue across days is a crucial task in neuroscience, which the paper is working on.
2. The paper as a whole is solid and all the components i.e., hierarchical alignment, causal architecture, used are sounded helpful for the stabilization task.
3. The experiments part are abundant and have a wide comparison across baselines and datasets.

**Weaknesses:**

1. The combination of the adversarial training and the hierarchical adaptation sounds tedious and a bit meaningless to me. If the proposed hierarchical domain adaptation is powerful enough, why do we still need the adversarial learning component. Thus, the contribution of the hierarchical adaptation is a bit weakened.
2. The theoretical verification is not strong enough since Lyapunov theory is actually a basic concept in control systems. Do you have more detailed and in-depth theory analysis on this part?
3. As shown in Line 277, the constraints added to the algorithm seems too many and no ablation study in the experiemtal part is provided.

**Questions:**

1. For the ERDiff baseline method [1], the reported results appear to differ from those originally presented. It also seems that CEBRA [2] is not a paper for the neural alignment task. Do you have a detailed re-run of ERDiff and CEBRA?

[1] Extraction and Recovery of Spatio-Temporal Structure in Latent Dynamics Alignment with Diffusion Models. NeurIPS 2023

[2] Learnable latent embeddings for joint behavioural and neural analysis. Nature 2023.

---

> ### Author Response · Authors · 2024-12-02
> **Response to Reviewer BT8N**
>
> Thanks for your constructive comments and insightful suggestions. We have gone through your review, and provided a point-by-point response to your questions.
>
> ### Weaknesses:
>
> 1. **The combination of the adversarial training and the hierarchical adaptation sounds tedious and a bit meaningless to me. If the proposed hierarchical domain adaptation is powerful enough, why do we still need the adversarial learning component. Thus, the contribution of the hierarchical adaptation is a bit weakened.**
>
>    Thank you for the comment. The effectiveness of hierarchical adaptation is demonstrated in Figure 2(a). We observed that the performance of HDA-r, which only includes latent space alignment, decreased significantly, highlighting the importance of alignment in the original space. Additionally, HDA-s (HDA without latent space alignment) performed worse than full HDA, particularly on the CO-M and RT-M datasets, which underscores the effectiveness of latent space alignment.
>
>    The adversarial learning component is employed for alignment in both the original space and latent space for HDA. As shown in Figure 7(b) on Page 21, adversarial alignment outperformed other approaches, such as MMD.
>
> 2. **The theoretical verification is not strong enough since Lyapunov theory is actually a basic concept in control systems. Do you have more detailed and in-depth theory analysis on this part?**
>
>    Thank you for the valuable feedback. In this work, the Lyapunov theory was mainly adopted as a metric to assess the stability of the model's cross-session performance, rather than providing new theoretical analysis. Thus, we did not focus on the theoretical part of Lyapunov theory. Using the Lyapunov theory as an evaluation metric, our results show that the HDA approach achieves improved dynamical stability after pre-alignment in the original space and self-consistent alignment in the latent space.
>
>    On the other hand, using the Lyapunov theory as a metric provides a novel approach  to evaluate the stability of a model’s cross-session performance, which has traditionally been assessed through supervised behavioral decoding across sessions.
>
> 3. **As shown in Line 277, the constraints added to the algorithm seems too many and no ablation study in the experimental part is provided.**
>
>    The constraints mentioned in Line 277 contribute to improving cross-session performance. Detailed ablation studies for each constraint are provided in the "Evaluation of Each Loss Term" (Section 4.3). As shown in Table 3, the performance decreased when $\lambda_y$, $\lambda_b$, and $\lambda_o$ were set to 0 respectively, demonstrating the effectiveness of these constraints.
>
> ### Questions:
>
> 1. **For the ERDiff baseline method, the reported results appear to differ from those originally presented. It also seems that CEBRA is not a paper for the neural alignment task. Do you have a detailed re-run of ERDiff and CEBRA?**
>
>    Thanks for your question. For ERDiff, we used the original code provided by authors but encountered vanishing gradient issues when applying it to our datasets. Upon further investigation, we found that this problem was likely related to the calculation of Sinkhorn Divergences. To address this, we refined the original calculation method, and obtained the results reported in our paper. Furthermore, our results were consistent with those reported in Table 2 ($R^2 = -0.32$) of [1].
>
>    For CEBRA, we selected this method to demonstrate the necessity of neural alignment. We also used the original code provided by authors and validated it on the target sessions without any alignment.
>
>    _[1] Vermani A, Park I M, Nassar J. Leveraging Generative Models for Unsupervised Alignment of Neural Time Series Data. In The Twelfth International Conference on Learning Representations, 2024._
>
> We sincerely appreciate your comments and suggestions, which are valuable for improving our work. Thanks for your time and consideration.

---

> > ### Comment · Reviewer_BT8N · 2024-12-02
> >
> > Thank you for your detailed response. I appreciate your discussions regarding the theoretical verification, which I find partially convincing. The addition of the "Evaluation of Each Loss Term" ablation study section is good and adds clarity to the methodology. However, the baseline comparisons still raise some concerns to me. It seems unusual that methods like Cycle-GAN and NoMAD consistently perform robustly across whole all different days, while others, such as LSTM, CEBRA, and ERDiff, consistently underperform, especially given that these methods do not differ fundamentally in their approach. Comparing your results to other studies like [1] seems less meaningful due to differences in experimental settings. For instance, while Cycle-GAN performs well in your results, it exhibits instability in Table 2 of [1]. Moreover, since the codebases for ERDiff and CEBRA have been updated since their initial release, re-running their experiments with the latest versions could enhance the reliability of your experimental validation. Addressing these points would further strengthen the paper, and I would be happy to consider revising my score once these concerns are resolved.
> >
> > [1] Vermani A, Park I M, Nassar J. Leveraging Generative Models for Unsupervised Alignment of Neural Time Series Data. In The Twelfth International Conference on Learning Representations, 2024.

---

> ### Author Response · Authors · 2024-12-02
>
> Thanks for your further feedback. To further investigate the unusual performance of CEBRA and ERDiff, we re-ran these two baselines using their latest code versions (CEBRA: version 0.4.0, ERDiff: updated 3 weeks ago).
>
> Their cross-performance results on the CO-M and RT-M datasets are presented in the two tables below. We found that the performance of CEBRA remains similar to the previous results. This may be because neither CEBRA nor LSTM employs the distribution alignment, unlike the other approaches. Consequently, CEBRA's performance is reasonable, given its lack of access to target data. For ERDiff, we observed an improvement compared to the previous results. The gradient instability issue has been solved in the latest version, and we will further refine its hyperparameters to achieve better performance in future work.
>
> Thanks once again for your time and consideration.
>
> #### Comparison of average $R^2$ scores (%) in cross-session velocity decoding on CO-M dataset
> |       | Day 8 | Day 14 | Day 15 | Day 22 | Day 24 | Day 25 | Day 28 | Day 29 | Day 31 | Day 32 |
> |-------|-------|--------|--------|--------|--------|--------|--------|--------|--------|--------|
> | CEBRA  | -115.06 ± 9.25 | -24.36 ± 17.07 | -127.53 ± 10.64 | -61.53 ± 12.91 | 8.94 ± 5.19 | -119.76 ± 9.89 | -90.17 ± 7.15 | -125.71 ± 9.30 | -121.58 ± 21.55 | -105.49 ± 20.96 |
> | ERDiff | -49.70 ± 65.16 | -33.59 ± 42.83 | -20.55 ± 40.02 | -33.08 ± 46.26 | -45.53 ± 62.31 | -68.93 ± 72.75 | -37.70 ± 54.12 | -26.44 ± 32.60 | -46.06 ± 57.27 | -59.85 ± 47.69 |
>
> #### Comparison of average $R^2$ scores (%) in cross-session velocity decoding on RT-M dataset
> |       | Day 1 | Day 38 | Day 39 | Day 40 | Day 52 | Day 53 | Day 67 | Day 69 | Day 77 | Day 79 |
> |-------|-------|--------|--------|--------|--------|--------|--------|--------|--------|--------|
> | CEBRA  | 67.63 ± 1.41 | 18.34 ± 10.11 | -16.94 ± 26.66 | -0.06 ± 23.70 | 26.86 ± 5.79 | 41.07 ± 5.42 | 29.53 ± 7.40 | -20.70 ± 34.13 | -59.47 ± 13.85 | -35.41 ± 18.91 |
> | ERDiff | -24.16 ± 43.76 | -21.76 ± 26.28 | -30.57 ± 36.75 | -59.25 ± 38.04 | -0.08 ± 0.06 | -42.03 ± 58.14 | -49.10 ± 27.76 | -22.09 ± 37.74 | -10.12 ± 17.87 | -10.06 ± 18.90 |

---

> > ### Comment · Reviewer_BT8N · 2024-12-02
> >
> > Thank you for conducting additional experiments on the baselines. While, from a practical perspective, it is somewhat intuitive that the baselines, after alignment, would achieve positive R-squared values, as they are expected to outperform the mean prediction. I hope the results, incorporating fair model tuning and hyperparameters, will be reflected in your updated work.

---

> > > ### Author Response · Authors · 2024-12-03
> > >
> > > We sincerely appreciate your further feedback and updating the score. Addditional results on fair model tuning and hyperparameters will be included in our updated work.

---

### Meta-Review · Area_Chair_63tG · 2024-12-18

**Metareview:**

The paper proposes a hierarchical domain adaptation (HDA) approach for neural alignment, aligning neural activity across multiple days by combining raw data and latent space alignment. Experimental results demonstrate improved behavior decoding accuracy compared to baselines, such as CEBRA and CycleGAN, supported by theoretical foundations from Lyapunov theory.

Strengths include clear experimental results and comparisons, a theoretical foundation, and addressing a critical challenge in neural alignment. However, the paper suffers from several significant weaknesses. The training strategy appears counterintuitive, as the model is trained simultaneously on source and target data rather than initially focusing on Day 0 data and aligning subsequent days. This raises questions about the framework's advantage over directly retraining models on new data, as the decoding performance does not consistently exceed simpler approaches. The necessity of accessing large amounts of Day 0 data during alignment limits real-world applicability. Furthermore, discrepancies in baseline performance (e.g., NoMAD and CEBRA) compared to previous studies undermine confidence in the results. The use of both GANs and VAEs appears redundant, and the alternating training process is inefficient. The writing and structure need significant improvement, with unclear motivations for certain design choices, overuse of mathematical notation, and insufficient comparisons to recent work. The claim of real-time BCI applicability is overstated due to the requirement of future day training data. Overall, these limitations outweigh the contributions, suggesting the paper is not yet ready for acceptance.

**Additional Comments On Reviewer Discussion:**

Reviewer comments highlighted concerns about redundancy in combining GANs and VAEs, inefficiencies in alternating network training, unclear loss term coefficients, and the choice of Chi-square divergence over KL divergence. Authors responded by demonstrating the necessity of GANs for cross-session performance, showing evidence of efficiency in alternating training, clarifying coefficients in Table 3, and justifying the stability benefits of Chi-square divergence. Questions regarding Lyapunov stability’s applicability to complex neural networks were acknowledged as requiring future validation. Another reviewer critiqued the daily retraining approach and reliance on Day 0 data, suggesting practical limitations; authors emphasized advantages in handling nonstationary neural activity and committed to future ablation studies. Errors in hyperparameter tuning for baseline methods and overstated claims about generalizability and independence were partially addressed with corrections and revisions. Despite these efforts, the absence of key ablation studies and incomplete empirical validations undermined confidence in the method, resulting in rejection.

---

### Decision · Program_Chairs · 2025-01-22

Reject